# Polynomial-Time Approximability of Constrained Reinforcement Learning

**Jeremy McMahan** [1]

## Abstract

We study the computational complexity of approximating general constrained Markov decision processes. Our primary contribution is the design of a polynomial time $(0, \epsilon)$-additive bicriteria approximation algorithm for finding optimal constrained policies across a broad class of recursively computable constraints, including almost-sure, chance, expectation, and their anytime variants. Matching lower bounds imply our approximation guarantees are optimal so long as $P \neq NP$. The generality of our approach results in answers to several long-standing open complexity questions in the constrained reinforcement learning literature. Specifically, we are the first to prove polynomial-time approximability for the following settings: policies under chance constraints, deterministic policies under multiple expectation constraints, policies under non-homogeneous constraints (i.e., constraints of different types), and policies under constraints for continuous-state processes.

## 1. Introduction

Constrained Reinforcement Learning (CRL) is growing increasingly crucial for managing complex, real-world applications such as medicine (Coronato et al., 2020; Paragliola et al., 2018; Kolesar, 1970), disaster relief (Fan et al., 2021; Wu et al., 2019; Tsai et al., 2019), and resource management (Mao et al., 2016; Li et al., 2018; Peng and Shen, 2021; Bhatia et al., 2021). Various constraints, including expectation (Altman, 1999), chance (Xu and Mannor, 2011), almost-sure (Castellano et al., 2022), and anytime constraints (McMahan and Zhu, 2024b), were each proposed to address new challenges. Despite the richness of the literature, most works focus on stochastic,

expectation-constrained policies, leaving many popular settings with longstanding open problems. Even chance constraints, arguably a close second in popularity, still lack any polynomial-time, even approximate, algorithms despite being introduced over a decade ago (Xu and Mannor, 2011). Other settings for which polynomial-time algorithms are open include deterministic policies under multiple expectation constraints, policies under non-homogeneous constraints (i.e., constraints of different types), and policies under constraints for continuous-state processes. Consequently, we study the computational complexity of general constrained problems to resolve many of these fundamental open questions.

Formally, we study the solution of *Constrained Markov Decision Processes* (CMDPs). Here, we define a CMDP through three fundamental parts: (1) an MDP $M$ that accumulates both rewards and costs, (2) a general cost criterion $C$, and (3) a budget vector $B$. Additionally, we allow the agent to specify whether they require their policy to be deterministic or stochastic, formalized through a goal policy class $\overline{\Pi}$. The agent's goal is to solve $\max_{\pi \in \overline{\Pi}} V_M^\pi$ subject to $C_M^\pi \leq B$, where $V_M^\pi$ denotes the agent's value and $C_M^\pi$ denotes the agent's cost under $\pi$. This model can capture very general problems, including minimum time routes for self-driving vehicles that must satisfy 1) an anytime constraint on fuel consumption, 2) an expectation constraint on $CO_2$ consumption, and 3) a chance constraint on vehicle wear and tear. Our main question is the following:

> Can general CMDPs be approximated in polynomial time?

**Hardness.** Solving general CMDPs is notoriously challenging. When restricted to deterministic policies, solving a CMDP with just one constraint is NP-hard (Feinberg, 2000; Xu and Mannor, 2011; McMahan and Zhu, 2024b; McMahan, 2024). This difficulty increases with the number of constraints: with at least two constraints, finding a feasible deterministic policy, let alone a near-optimal one, becomes NP-hard (McMahan and Zhu, 2024b). Even if we relax the deterministic requirement, this hardness remains for all constraint types other than expectation. Computational hardness aside, standard RL techniques fail to apply due to the combinatorial nature induced by many constraint

---

[1]Department of Computer Science, University of Wisconsin-Madison, Wisconsin, USA. Correspondence to: Jeremy McMahan <jmcmahan@wisc.edu>.

*Proceedings of the $42^{nd}$ International Conference on Machine Learning*, Vancouver, Canada. PMLR 267, 2025. Copyright 2025 by the author(s).

types. Adding in additional constraints with fundamentally different structures further complicates the problem.

**Past Work.** A few works have managed to derive provable approximation algorithms for some cases of CRL. McMahan (2024) presented a fully polynomial-time approximation scheme (FPTAS) for the computation of deterministic policies of a general class of constraints, which includes expectation, almost-sure, and anytime constraints. Although powerful, their framework only works for one constraint and fails to capture anytime-expectation constraints along with chance constraints. Similarly, Khonji et al. (2019) achieves an FPTAS for expectation and chance constraints, but only in the constant horizon setting. In contrast, McMahan and Zhu (2024b) develops a polynomial-time $(0, \epsilon)$-additive bicritiera approximation algorithm for anytime and almost-sure constraints. However, their framework is specialized to those constraint types and thus fails for our purpose. In contrast, Xu and Mannor (2011) developed a pseudo-polynomial time algorithm for finding feasible chance-constrained policies, but their methods do not lead to polynomial-time solutions.

**Our Contributions.** We design a polynomial-time $(0, \epsilon)$-additive bicriteria approximation algorithm for tabular, SR-criterion CMDPs. An SR criterion is required to satisfy a generalization of the policy evaluation equations and includes expectation, chance, and almost-sure constraints along with their anytime equivalents. Our framework implies the first positive polynomial-time approximability results for (1) policies under chance constraints, (2) deterministic policies under multiple expectation constraints, and (3) policies under non-homogeneous constraints – each of which has been unresolved for over a decade. We then extend our algorithm into a polynomial-time $(\epsilon, \epsilon)$-additive bicriteria approximation algorithm for continuous-state CMPDs under a general class of constraints, which includes expectation, almost-sure, and anytime constraints.

**Our Techniques.** Our algorithm requires several key techniques. First, we transform a constraint concerning all realizable histories into a simpler per-time constraint. We accomplish this by augmenting the state space with an artificial budget and augmenting the action space to choose future budgets to satisfy the constraint. However, Bellman updates then become as hard as the knapsack problem due to the large augmented action space. For tabular cMDPs, we show that the Bellman updates can be approximately computed using dynamic programming and rounding. By strategically rounding the artificial budget space, we achieve a $(0, \epsilon)$-bicriteria approximation for tabular CMDPs. By appropriately discretizing the continuous state space, our method becomes a $(\epsilon, \epsilon)$-bicriteria approximation algorithm for continuous state CMDPs.

### 1.1. Related Work

**Constrained RL.** It is known that stochastic expectation-constrained policies are polynomial-time computable via linear programming (Altman, 1999), and many planning and learning algorithms exist for them (Paternain et al., 2019; Vaswani et al., 2022; Borkar, 2005; HasanzadeZonuzy et al., 2021). Some learning algorithms can even avoid violation during the learning process under certain assumptions (Wei et al., 2022; Bai et al., 2023). Furthermore, Brantley et al. (2020) developed no-regret algorithms for cMDPs and extended their algorithms to the setting with a constraint on the cost accumulated over all episodes, which is called a knapsack constraint (Brantley et al., 2020; Cheung, 2019).

**Safe RL.** The safe RL community (García et al., 2015; Gu et al., 2024) has mainly focused on no-violation learning for stochastic expectation-constrained policies (Chow et al., 2018; Bossens and Bishop, 2022; Alshiekh et al., 2018; Cheng et al., 2019; Berkenkamp et al., 2017) and solving chance constraints (Wang et al., 2023; Zhao et al., 2023), which capture the probability of entering unsafe states. Performing learning while avoiding dangerous states (Zhao et al., 2023) is a special case of expectation constraints that has also been studied (Roderick et al., 2021; Thomas et al., 2021) under non-trivial assumptions. In addition, instantaneous constraints, which require the immediate cost to be within budget at all times, have also been studied (Li et al., 2021; Fisac et al., 2019; Gros et al., 2020).

## 2. Constraints

**Cost-Accumulating MDPs.** In this work, we consider environments that accumulate both rewards and costs. Formally, we consider a (finite-horizon, tabular) *cost-accumulating Markov Decision Process* (caMDP) tuple $M = (H, \mathcal{S}, \mathcal{A}, P, R, C, s_0)$, where (i) $H \in \mathbb{Z}_{\geq 0}$ is the finite time *horizon*, (ii) $\mathcal{S}_h$ is the finite set of *states*, (iii) $\mathcal{A}_h(s)$ is the finite set of available *actions*, (iv) $P_h(s, a) \in \Delta(S)$ is the *transition* distribution for a given state $s \in \mathcal{S}$ and action $a \in \mathcal{A}$ (note, $\Delta(S)$ represents the probability simplex over $\mathcal{S}$), (v) $R_h(s, a) \in \Delta(\mathbb{R})$ is the *reward* distribution, (vi) $C_h(s, a) \in \Delta(\mathbb{R}^m)$ is the *cost* distribution, and (vii) $s_0 \in \mathcal{S}$ is the initial state.

To simplify notation, we let $r_h(s, a) \stackrel{\text{def}}{=} \mathbb{E}[R_h(s, a)]$ denote the expected reward, $c_h(s, a)$ represent the cost function when costs are deterministic (which will be the case throughout the main text), $S \stackrel{\text{def}}{=} |\mathcal{S}|$ denote the number of states, $A \stackrel{\text{def}}{=} |\mathcal{A}|$ denote the number of joint actions, $[H] \stackrel{\text{def}}{=} \{1, \ldots, H\}$, $\mathcal{M}$ be the set of all caMDPs, and $|M|$ be the total description size of the caMDP. We also use the Iverson Bracket notation $[P] \stackrel{\text{def}}{=} 1_{\{P=True\}}$ and the characteristic function $\chi_P$ which is $\infty$ when $P$ is False and 0

otherwise.

**Agent Interactions.** The agent interacts with $M$ using a *policy* $\pi = \{\pi_h\}_{h=1}^H$. In the fullest generality, $\pi_h : \mathcal{H}_h \to \Delta(\mathcal{A})$ is a mapping from the observed history at time $h$ (including costs) to a distribution of actions. Often, researchers study *Markovian policies*, which take the form $\pi_h : \mathcal{S} \to \Delta(\mathcal{A})$, and *deterministic policies*, which take the form $\pi_h : \mathcal{H}_h \to \mathcal{A}$. We let $\Pi$ denote the set of all policies and $\Pi^D$ denote the set of all deterministic policies.

The agent starts in the initial state $s_0$ with observed history $\tau_1 = (s_0)$. For any $h \in [H]$, the agent chooses a joint action $a_h \sim \pi_h(\tau_h)$. Then, the agent receives immediate reward $r_h \sim R_h(s, a)$ and cost vector $c_h \sim C_h(s, a)$. Lastly, $M$ transitions to state $s_{h+1} \sim P_h(s_h, a_h)$, prompting the agent to update its observed history to $\tau_{h+1} = (\tau_h, a_h, c_h, s_{h+1})$. This process is repeated for $H$ steps; the interaction ends once $s_{H+1}$ is reached.

**Constrained Processes.** Suppose the agent has a *cost criterion* $C : \mathcal{M} \times \Pi \to \mathbb{R}^m$ and a corresponding *budget* vector $B \in \mathbb{R}^m$. We refer to the tuple $(M, C, B)$ as a *Constrained Markov Decision Process* (CMDP). Given a CMDP and desired policy class $\overline{\Pi} \in \{\Pi^D, \Pi\}$, the agent's goal is to solve the constrained optimization problem:

$$\max_{\pi \in \overline{\Pi}} \ V_M^\pi \tag{CON}$$
$$\text{s.t. } C_M^\pi \leq B$$

In the above, $V_M^\pi \overset{\text{def}}{=} \mathbb{E}_M^\pi \left[ \sum_{h=1}^H r_h(s_h, a_h) \right]$ denotes the value of a policy $\pi$, $\mathbb{E}_M^\pi$ denotes the expectation defined by $\mathbb{P}_M^\pi$, and $\mathbb{P}_M^\pi$ denotes the probability law over histories induced from the interaction of $\pi$ with $M$. Lastly, we let $V^*$ denote the optimal solution value to (CON). In the main paper, we focus on the case where $\overline{\Pi} = \Pi^D$.

**SR Criteria.** We study cost criteria that generalize the standard policy evaluation equations to enable dynamic programming techniques. In particular, we require the cost of a policy to be recursively computable with respect to the time horizon. For our later approximations in Section 5, we will also need key functions defining the recursion to be short maps, i.e., 1-Lipschitz, with respect to the infinity norm.

**Definition 2.1** (SR). We call a cost criterion *shortly recursive* (SR) if for any caMDP $M$ and any policy $\pi \in \Pi^D$, $\pi$'s cost decomposes recursively into $C_M^\pi = C_1^\pi(s_0)$, where $C_{H+1}^\pi = 0$ and for all $h \in [H]$ and $\tau_h \in \mathcal{H}_h$ letting $s = s_h(\tau_h)$ and $a = \pi_h(\tau_h)$,

$$C_h^\pi(\tau_h) = c_h(s, a) + \underset{s'}{f} \ g \left( P_h(s' \mid s, a) \right) C_{h+1}^\pi(\tau_h, a, s'). \tag{SR}$$

Here, $f_{s'}$ is the finite extension of an associative, nondecreasing, binary function $f$, and $g$ is a $[0, 1]$-valued

function rooted at $0$. Moreover, we require that $f$ is a short map when either of its inputs are fixed, satisfies $f(0, x) = f(x, 0) = x$ for all $x$, and when combined with $g$, i.e., $f_{s'} \ g \left( P_h(s' \mid s, a) \right) x_{s'}$, is a short map in $x$.

*Remark* 2.2 (Stochastic Variants). Our results generalize to both stochastic policies and stochastic costs as well. The algorithmic approach is identical, but the definitions and analysis are more complex. Consequently, we focus on the deterministic cases in the main text.

**Constraint Formulations.** The fundamental constraints considered in the CRL literature are Expectation, Chance, and Almost-sure constraints. Each of these induces a natural *anytime* variant that stipulates the required constraint must hold for the truncated cost $\sum_{h=1}^t c_h$ at all times $h \in [H]$. We give the formal definitions in Figure 1. Importantly, each constraint is equivalent to $C_M^\pi \leq B'$ for some appropriately chosen SR criteria.

**Proposition 2.3** (SR Modeling). *The classical constraints can be modeled by SR constraints of the form $C_M^\pi \leq B'$ as follows:*

1. *Expectation Constraints –* $f(x, y) \overset{\text{def}}{=} x + y$, $g(x) \overset{\text{def}}{=} x$, *and* $B' \overset{\text{def}}{=} B$.

2. *Chance Constraints –* $(f, g)$ *defined as above and* $B' \overset{\text{def}}{=} \delta$. *Here, we assume $M$'s states are augmented with cumulative costs and that* $c_h((s, \bar{c}), a) \overset{\text{def}}{=} [c_h(s, a) + \bar{c} > B]$ *for the anytime variant and* $c_h((s, \bar{c}), a) \overset{\text{def}}{=} [c_h(s, a) + \bar{c} > B][h = H]$ *otherwise.*

3. *Almost-sure Constraints –* $f(x, y) \overset{\text{def}}{=} \max(x, y)$, $g(x) \overset{\text{def}}{=} [x > 0]$, *and* $B' \overset{\text{def}}{=} B$. *Anytime variant –* $f(x, y) \overset{\text{def}}{=} \max(0, \max(x, y))$ *while $g$ and $B'$ remain the same.*

*General anytime variants, including anytime expectation constraints, can be modeled by* $\left\{ C_{M,t}^\pi \leq B \right\}_{t \in [H]}$ *where $C_{M,t}^\pi$ is the original SR criterion but defined for the truncated-horizon process with horizon $t$.*

**Computational Limitations.** It is known that computing feasible policies for CMDPs is NP-hard (McMahan, 2024; McMahan and Zhu, 2024b). As such, we must relax feasibility for any hope of polynomial-time algorithms. Consequently, we focus on *bicriteria* approximation algorithms.

**Definition 2.4** (Bicriteria). A policy $\pi$ is an $(\alpha, \beta)$-*additive bicriteria approximation* to a CMDP $(M, C, B)$ if $V_M^\pi \geq V^* - \alpha$ and $C_M^\pi \leq B + \beta$. We refer to an algorithm as an $(\alpha, \beta)$-*bicriteria* if for any CMDP it outputs an $(\alpha, \beta)$-*additive bicriteria approximation* or correctly reports the instance is infeasible.

| Con/Time | Expectation | Chance | Almost-Sure |
|---|---|---|---|
| Classical | $\mathbb{E}_M^\pi \left[ \sum_{h=1}^{H} c_h \right] \le B$ | $\mathbb{P}_M^\pi \left[ \sum_{h=1}^{H} c_h > B \right] \le \delta$ | $\mathbb{P}_M^\pi \left[ \sum_{h=1}^{H} c_h \le B \right] = 1$ |
| Anytime ($\forall t \in [H]$) | $\mathbb{E}_M^\pi \left[ \sum_{h=1}^{t} c_h \right] \le B$ | $\mathbb{P}_M^\pi \left[ \sum_{h=1}^{t} c_h > B \right] \le \delta$ | $\mathbb{P}_M^\pi \left[ \sum_{h=1}^{t} c_h \le B \right] = 1$ |

*Figure 1.* The Constraint Landscape

The existence of a polynomial-time bicriterion for our general constrained problem implies brand-new approximability results for many classic problems in the CRL literature. For clarity, we will explicitly state the complexity-theoretic implications for each classical setting.

**Theorem 2.5** (Implications). *A polynomial-time $(\epsilon, \epsilon)$-bicriteria implies that in polynomial time it is possible to compute a policy $\pi \in \overline{\Pi}$ satisfying $V_M^\pi \ge V^* - \epsilon$ and any constant combination of the following constraints:*

1. $\mathbb{E}_M^\pi \left[ \sum_{h=1}^{H} c_h \right] \le B + \epsilon$

2. $\mathbb{P}_M^\pi \left[ \sum_{h=1}^{H} c_h \le B + \epsilon \right] = 1$

3. $\mathbb{P}_M^\pi \left[ \sum_{h=1}^{H} c_h > B + \epsilon \right] \le \delta + \epsilon.$

*In other words, polynomial-time approximability is possible for each of the settings described in Section 1 when the number of constraints is constant.*

*Remark* 2.6 (Extensions). All of our results hold for Markov games and infinite discounted settings.

## 3. Reduction

In this section, we present a general solution approach to SR-criterion CMDPs. Our approach revolves around converting the general cost constraint into a per-step action constraint. This leads to the design of an augmented MDP that can be solved with standard RL methods.

**Augmentation.** State augmentation is the known approach for solving anytime-constrained MDPs (McMahan and Zhu, 2024b). The augmentation permits the problem to be solved by the following dynamic program:

$$V_h^*(s, c) = \max_{\substack{a \in \mathcal{A}, \\ c + c_h(s,a) \le B}} r_h(s, a)$$
$$+ \sum_{s'} P_h(s' \mid s, a) V_{h+1}^*(s, c + c_h(s, a)). \quad (1)$$

When moving to other constraints, the cumulative cost may no longer suffice to determine constraint satisfaction. For example, the expected cost depends on the cumulative cost of all realizable branches, not just the current branch.

**Expectation Constraints.** Instead, we can exploit the recursive nature of the expected cost to find a solution. Suppose at stage $(s, h)$ we impose an artificial budget $b$ on the expected cost of a policy $\pi$ from time $h$ onward: $\mathbb{E}^\pi \left[ \sum_{t=h}^{H} c_t \right] \le b$. By the policy evaluation equations, we know this equates to satisfying:

$$C_h^\pi(s) = c_h(s, a) + \sum_{s'} P_h(s' \mid s, a) C_{h+1}^\pi(s') \le b. \quad (2)$$

For this invariant to be satisfied, it suffices for the agent to choose future artificial budgets $b_{s'}$ for $s' \in \mathcal{S}$ satisfying,

$$c_h(s, a) + \sum_{s'} P_h(s' \mid s, a) b_{s'} \le b. \quad (3)$$

and ensure the future artificial budgets are obeyed inductively: $C_{h+1}^\pi(s', b_{s'}) \le b_{s'}$.

**General Approach.** We can apply the same reasoning for general recursively computable cost criteria. If $C$ is SR, then we know that $C_h^\pi(s)$ obeys (SR). Thus, to guarantee that $C_h^\pi(s) \le b$ it suffices to choose $b_{s'}$'s satisfying,

$$c_h(s, a) + \underset{s'}{f} \, g \left( P_h(s' \mid s, a) \right) b_{s'} \le b, \quad (4)$$

and inductively guarantee that $C_{h+1}^\pi(s') \le b_{s'}$.

We can view choosing future artificial budgets as part of the agent's augmented actions. Then, at any augmented state $(s, b)$, the agent's augmented action space includes all $(a, \mathbf{b}) \in \mathcal{A} \times \mathbb{R}^S$ satisfying (3). When $M$ transitions to $s' \sim P_h(s, a)$, the agent's new augmented state should consist of the environment's new state in addition to its chosen demand for that state, $(s', b_{s'})$. Putting these pieces together yields the definition of the reduced, action-constrained MDP, Definition 3.1.

**Definition 3.1** (Reduced MDP). Given any SR-criterion CMDP $(M, C, B)$, we define the *reduced MDP* $\bar{M} \overset{\text{def}}{=} (H, \bar{\mathcal{S}}, \bar{\mathcal{A}}, \bar{P}, \bar{R}, \bar{s}_0)$ where,

1. $\bar{\mathcal{S}}_h \overset{\text{def}}{=} \mathcal{S}_h \times \mathcal{B}$ where $\mathcal{B} \overset{\text{def}}{=} \bigcup_{\pi \in \Pi^D} \bigcup_{h \in [H+1]} \bigcup_{\tau_h \in \mathcal{H}_h} \{ C_h^\pi(\tau_h) \}$

2. $\bar{\mathcal{A}}_h(s, b) \overset{\text{def}}{=} \{ (a, \mathbf{b}) \in \mathcal{A}_h(s) \times \mathbb{R}^S \mid c_h(s, a) + \underset{s'}{f} \, g \left( P_h(s' \mid s, a), b_{s'} \right) \le b \}$

**Algorithm 1** Reduction

**Require:** $(M, C, B)$
 1: $\bar{M} \leftarrow$ Definition 3.1$(M, C, B)$
 2: $\pi, \bar{V}^* \leftarrow$ SOLVE$(\bar{M})$
 3: **if** $\bar{V}^* = -\infty$ **then**
 4:     **return** "Infeasible"
 5: **else**
 6:     **return** $\pi$
 7: **end if**

**Algorithm 2** Augmented Interaction

**Require:** $\pi$
 1: $\bar{s}_1 = (s_0, B)$
 2: **for** $h \leftarrow 1$ to $H$ **do**
 3:     $(a, \mathbf{b}) \leftarrow \pi_h(\bar{s}_h)$
 4:     $s_{h+1} \sim P_h(s_h, a)$
 5:     $\bar{s}_{h+1} = (s_{h+1}, b_{s_{h+1}})$
 6: **end for**

3. $\bar{P}_h((s', b') \mid (s, b), (a, \mathbf{b})) \stackrel{\text{def}}{=} P_h(s' \mid s, a)[b' = b_{s'}]$

4. $\overline{R}_h((s, b), (a, \mathbf{b})) \stackrel{\text{def}}{=} R_h(s, a)$

5. $\bar{s}_0 \stackrel{\text{def}}{=} (s_0, B)$

We also re-define the base case value to $\bar{V}^*_{H+1}(s, b) \stackrel{\text{def}}{=} -\chi_{\{b \geq 0\}}$. Note, the reduced MDP has a non-stationary state and action set, unlike the base MDP.

**Reduction.** Importantly, $\bar{M}$'s augmented action space ensures constraint satisfaction. Thus, we have effectively reduced a problem involving total history constraints to one with only standard per-time-step constraints. So long as our cost is SR, $\bar{M}$ can be solved using fast RL methods instead of the brute force computation required for general CMDPs. These properties ensure our method, Algorithm 1, is correct.

**Lemma 3.2** (Value). *For any $h \in [H+1]$, $\tau_h \in \mathcal{H}_h$, and $b \in \mathcal{B}$, if $s = s_h(\tau_h)$, then,*

$$\bar{V}^*_h(s, b) \geq \sup_{\pi \in \Pi^D} V^\pi_h(\tau_h) \qquad (5)$$
$$s.t. \ C^\pi_h(\tau_h) \leq b.$$

**Lemma 3.3** (Cost). *Suppose that $\pi \in \Pi^D$. For all $h \in [H+1]$ and $(s, b) \in \bar{\mathcal{S}}$, if $\bar{V}^\pi_h(s, b) > -\infty$, then $\bar{C}^\pi_h(s, b) \leq b$.*

**Theorem 3.4** (Reduction). *If SOLVE is any finite-time MDP solver, then Algorithm 1 correctly solves (CON) in finite time for any SR-criterion CMDP.*

*Remark* 3.5 (Deployment). Given a budget-augmented policy $\pi$ output from Algorithm 1, the agent can execute $\pi$ using Algorithm 2.

## 4. Bellman Updates

In this section, we discuss efficient methods for solving $\bar{M}$. Our approach relies on using (SR) to break down the Bellman update so that it is solvable using dynamic programming. We then use dynamic rounding to achieve an efficient approximation algorithm.

**Bellman Hardness.** Even a small set of artificial budgets, $\mathcal{B}$, needed to be considered, solving $\bar{M}$ would still be challenging due to its exponentially large, constrained action space. Just one Bellman update equates to solving the constrained optimization problem:

$$\bar{V}^*_h(s, b) = \max_{a, \mathbf{b}} r_h(s, a) + \sum_{s'} P_h(s' \mid s, a) V^*_{h+1}(s', b_{s'})$$
$$\text{s.t. } c_h(s, a) + \underset{s'}{f} g(P_h(s' \mid s, a)) b_{s'} \leq b.$$
$$(\text{BU})$$

Above, we used the fact that $(s', b') \in$ Supp$(\bar{P}_h((s, b), (a, \mathbf{b})))$ iff $s' \in$ Supp$(P_h(s, a))$ and $b' = b_{s'}$. In fact, even when each $b_{s'}$ only takes on two possible values, $\{0, w_{s'}\}$, this optimization problem generalizes the knapsack problem, implying that it is NP-hard to solve.

**Dynamic Programming.** To get around this computational bottleneck, we must fully exploit Definition 2.1. For any fixed $(h, (s, b), a)$, the key idea is to treat choosing $b'$s as its own sequential decision-making problem. Suppose we have already chosen $b_1, \ldots, b_{t-1}$ leading to partial cost $F \stackrel{\text{def}}{=} f^{t-1}_{s'=1} g(P_h(s' \mid s, a)) b_{s'}$. Since $f$ is associative, we can update our partial cost after choosing $b_t$ to $f(F, g(P_h(t \mid s, a)) b_t)$. Once we have made a choice for each future state, we can verify if $(a, \mathbf{b}) \in \bar{\mathcal{A}}_h(s, b)$ by checking the condition: $c_h(s, a) + F \leq b$. By incorporating the value objective, we design a dynamic program for computing (BU).

**Definition 4.1** (DP). *For any $h \in [H]$, $(s, b) \in \bar{\mathcal{S}}$, $a \in \mathcal{A}$ and $F \in \mathbb{R}$, we define $\bar{V}^{s,a}_{h,b}(S+1, F) = -\chi_{\{c_h(s,a)+F \leq b\}}$, and for any $t \in [S]$,*

$$\bar{V}^{s,a}_{h,b}(t, F) \stackrel{\text{def}}{=} \max_{b_t \in \mathcal{B}} P_h(t \mid s, a) \bar{V}^*_{h+1}(t, b_t) +$$
$$\bar{V}^{s,a}_{h,b}(t+1, f(F, g(P_h(t \mid s, a)) b_t)). \qquad (6)$$

**Lemma 4.2** (DP Correctness). *For any $h \in [H]$ and $(s, b) \in \bar{\mathcal{S}}$, we have that $\bar{V}^*_h(s, b) = \max_{a \in \mathcal{A}} r_h(s, a) + \bar{V}^{s,a}_{h,b}(1, 0)$.*

**Dynamic Rounding.** Although a step in the right direction, solving Definition 4.1 can still be slow due to the exponential number of considered partial costs. We resolve this issue by rounding each partial cost to an element of some

small set $\hat{\mathcal{F}}$. Since $f$ need not be linear, using rounding in a preprocessing step does not suffice: we must re-round at each step to ensure inputs are a valid element of our input set.

For any $\ell > 0$, we view $\ell$ as a new unit length. Our rounding function maps any real number to its closest upper bound in the set of integer multiples of $\ell$. We use upper bounds to guarantee that the rounded partial costs are always larger than the true partial costs. Smaller $\ell$ ensures less approximation error, while larger $\ell$ ensures fewer considered partial costs. Thus, $\ell$ directly controls the accuracy-efficiency trade-off.

**Definition 4.3** (Rounding Functions). For any $\ell > 0$ and $x \in \mathbb{R}$, we define $\lceil x \rceil_\ell \overset{\text{def}}{=} \lceil \frac{x}{\ell} \rceil \ell$ to be the smallest integer multiple of $\ell$ that is larger than $x$. We also define $\kappa_\ell(x) \overset{\text{def}}{=} x + \ell(S+1)$. Note, when considering vectors, all operations are performed component-wise.

Since we round up the partial costs, the approximate partial cost of a feasible $\mathbf{b}$ could exceed $b$. To ensure all feasible choices of $\mathbf{b}$ are considered, we must also relax the budget comparison. Instead, we compare partial costs to a carefully chosen upper threshold $\kappa(b)$. Putting these pieces together yields our approximate Bellman update method.

**Definition 4.4** (Approximate Update). Fix any $\ell > 0$ and function $\kappa : \mathbb{R}^m \to \mathbb{R}^m$. For any $h \in [H]$, $(s,b) \in \bar{\mathcal{S}}$, $a \in \mathcal{A}$ and $\hat{F} \in \mathbb{R}^m$, we define $\hat{V}_{h,b}^{s,a}(S+1, \hat{F}) \overset{\text{def}}{=} -\chi_{\{c_h(s,a) + \hat{F} \le \kappa(b)\}}$, and for any $t \in [S]$,

$$
\begin{aligned}
\hat{V}_{h,b}^{s,a}(t, \hat{F}) &\overset{\text{def}}{=} \max_{b_t \in \mathcal{B}} P_h(t \mid s,a) \bar{V}_{h+1}^*(t, b_t) + \\
&\hat{V}_{h,b}^{s,a}\left(t+1, \left\lceil f\left(\hat{F}, g(P_h(t \mid s,a)) b_t\right) \right\rceil_\ell\right).
\end{aligned}
\tag{ADP}
$$

We then define the *approximate update* by,

$$
\hat{V}_h^*(s,b) \overset{\text{def}}{=} \max_{a \in \mathcal{A}} r_h(s,a) + \hat{V}_{h,b}^{s,a}(1, 0).
\tag{AU}
$$

Overall, solving the ADP yields an approximate solution.

**Lemma 4.5** (Approximation). *For any $h \in [H]$, $(s,b) \in \bar{\mathcal{S}}$, $a \in \mathcal{A}$, $\hat{F} \in \mathbb{R}^m$, and $t \in [S+1]$, we have that,*

$$
\begin{aligned}
\hat{V}_{h,b}^{s,a}(t, \hat{F}) = \max_{\mathbf{b} \in \mathcal{B}^{S-t+1}} &\sum_{s'=t}^{S} P_h(s' \mid s,a) \bar{V}_{h+1}^*(s', b_{s'}) \\
s.t. \quad &c_h(s,a) + \hat{f}_{h,\mathbf{b}}^{s,a}(t, \hat{F}) \le \kappa(b),
\end{aligned}
\tag{7}
$$

*where $\hat{f}_{h,\mathbf{b}}^{s,a}(t, \hat{F})$ is the dynamic rounding of $f\left(\hat{F}, f_{s'=t}^S g(P_h(t \mid s,a), b_t)\right)$. Moreover, if $\lceil \rceil_\ell$ and $\kappa$ are replaced with the identity function, (AU) is equivalent to (BU).*

---

**Algorithm 3** Approximate Backward Induction

**Require:** $\bar{M}$
1: $\hat{V}_{H+1}^*(s,b) \leftarrow \chi_{\{b \ge 0\}}$ for all $(s,b) \in \bar{\mathcal{S}}$
2: **for** $h \leftarrow H$ down to $1$ **do**
3:     **for** $(s,b) \in \bar{\mathcal{S}}$ **do**
4:         $\hat{a}, \hat{V}_h^*(s,b) \leftarrow$ (AU)
5:         $\pi_h(s,b) \leftarrow \hat{a}$
6:     **end for**
7: **end for**
8: **return** $\pi, \hat{V}^*$

---

*Remark* 4.6 (DP details). Technically, to turn this recursion into a true dynamic program, we must also precompute the inputs to any subproblem. Unlike in standard RL, this computation must be done with a forward recursion. If we let $\hat{\mathcal{F}}_h^{s,a}(t)$ denote the set of possible input rounded partial costs for state $t$, then the set satisfies the inductive relationship $\hat{\mathcal{F}}_h^{s,a}(1) \overset{\text{def}}{=} \{0\}$ and for any $t \in [S]$, $\hat{\mathcal{F}}_h^{s,a}(t+1) \overset{\text{def}}{=} \bigcup_{b_t \in \mathcal{B}} \bigcup_{F \in \hat{\mathcal{F}}_h^{s,a}(t)} \{\lceil f(F, + g(P_h(t \mid s,a)) b_t \rceil_\ell\}$. This relationship translates directly into an iterative algorithm for computing all needed inputs. Using this gives a complete DP algorithm for solving (ADP)[1].

**Theorem 4.7** (Approx Solve). *When $\lceil \rceil_\ell$ and $\kappa$ are replaced with the identity function, Algorithm 3 correctly solves any $\bar{M}$ produced from Definition 3.1. Moreover, Algorithm 3 runs in time $O\left(H^{m+1} S^{m+2} A |\mathcal{B}|^2 \|c_{max} - c_{min}\|_\infty^m / \ell^m\right)$.*

## 5. Bicriteria

Algorithm 3 allows us to approximately solve $\bar{M}$ in finite cases much faster than traditional methods. However, when $|\mathcal{B}|$ is large, the algorithm still runs in exponential time. Similarly to the partial cost rounding in Definition 4.4, we can reduce the size of $|\mathcal{B}|$ by considering a smaller approximate set based on rounding. Since we still desire optimistic budgets, we use the same rounding function from Definition 4.3 but with a different choice of $\ell$.

**Budget Rounding.** Rounding naturally impacts the state space, but has other consequences as well. To avoid complex computation, we consider the approximate set $\hat{\mathcal{B}} \overset{\text{def}}{=} \{\lceil b \rceil_\ell \mid b \in [b_{min}, b_{max}]\}$ where $[b_{min}, b_{max}] \supseteq \mathcal{B}$ is a superset of all required artificial budgets that we formalize later. As before, rounding the budgets may cause originally feasible choices to now violate the constraint. To ensure all feasible choices are considered and that we can use Algorithm 3 to get speed-ups, we define the approximate action space to include all vectors that lead to feasible subproblems

---

[1] We use the notation $x, o \leftarrow \min_x z(x)$ to say that $x$ is the minimizer and $o$ the value of the optimization.

---

**Algorithm 4** Bicriteria

---

**Require:** $(M, C, B)$
1: **Hyperparameter:** $\ell$
2: $\hat{M} \leftarrow$ Definition 5.1$(M, (f, g), B, \ell)$
3: $\pi, \hat{V}^* \leftarrow$ Algorithm 3$(\hat{M}, (f, g), \ell)$
4: **if** $\hat{V}_1^*(s_0, \lceil B \rceil_\ell) = -\infty$ **then**
5:     **return** "Infeasible"
6: **else**
7:     **return** $\pi$
8: **end if**

---

of (ADP). From Lemma 4.5, we know this set is exactly the set of $(a, \hat{\mathbf{b}})$ satisfying $c_h(s, a) + \hat{f}_{h, \hat{\mathbf{b}}}^{s,a}(1, 0) \leq \kappa(\hat{b})$. Putting these ideas together yields a new, approximate MDP.

**Definition 5.1** (Approximate MDP). Given any SR-criterion CMDP $(M, C, B)$, we define the *approximate MDP* $\hat{M} \stackrel{\text{def}}{=} (H, \hat{\mathcal{S}}, \hat{\mathcal{A}}, \hat{P}, \hat{R}, \hat{s}_0)$ where,

1. $\hat{\mathcal{S}}_h \stackrel{\text{def}}{=} \mathcal{S}_h \times \hat{\mathcal{B}}$ where $\hat{\mathcal{B}} \stackrel{\text{def}}{=} \{\lceil b \rceil_\ell \mid b \in [b_{min}, b_{max}]\}$.

2. $\hat{\mathcal{A}}_h(s, \hat{b}) \stackrel{\text{def}}{=} \{(a, \hat{\mathbf{b}}) \in \mathcal{A}_h(s) \times \hat{\mathcal{B}}^S \mid c_h(s, a) + \hat{f}_{h, \hat{\mathbf{b}}}^{s,a}(1, 0) \leq \kappa(\hat{b})\}$

3. $\hat{P}_h((s', \hat{b}') \mid (s, b), (a, \hat{\mathbf{b}})) \stackrel{\text{def}}{=} P_h(s' \mid s, a)[\hat{b}' = \hat{b}_{s'}]$

4. $\hat{R}_h((s, \hat{b}), a) \stackrel{\text{def}}{=} R_h(s, a)$

5. $\hat{s}_0 \stackrel{\text{def}}{=} (s_0, \lceil B \rceil_\ell)$

We again re-define the base case value to $\hat{V}_{H+1}^*(s, \hat{b}) \stackrel{\text{def}}{=} -\chi_{\{\hat{b} \geq 0\}}$.

Since we always round budgets up, the agent can make even better choices than originally. It is then easy to see that policies for $\hat{M}$ always achieve optimal constrained value. We formalize this observation in Lemma 5.2.

**Lemma 5.2** (Optimal Value). *For any $h \in [H+1]$ and $(s, b) \in \bar{\mathcal{S}}$, $\hat{V}_h^*(s, \lceil b \rceil_\ell) \geq \bar{V}_h^*(s, b)$.*

**Time-Space Errors.** To assess the violation gap of Algorithm 4 policies, we must first explore the error accumulated by our rounding approach. Rounding each artificial budget naturally accumulates approximation error over time. Rounding the partial costs while running Algorithm 3 accumulates additional error over (state) space. Thus, solving $\hat{M}$ using Algorithm 4 accumulates error over both time and space, unlike standard approximate methods in RL. As a result, our rounding and threshold functions will generally depend on both $H$ and $S$.

**Arithmetic Rounding.** Our approach is to round each value down to its closest element in an $\ell$-cover. Using the same rounding as in Definition 4.3, we guarantee that $b \leq \lceil b \rceil_\ell \leq b + \ell$. Thus, $\lceil b \rceil_\ell$ is an overestimate that is not too far from the true value. By setting $\ell$ to be inversely proportional to $SH$, we control the errors over time and space. The lower bound must also be a function of $S$ since it controls the error over space.

**Lemma 5.3** (Approximate Cost). *Suppose that $\pi \in \Pi^D$. For all $h \in [H+1]$ and $(s, \hat{b}) \in \hat{\mathcal{S}}$, if $\hat{V}_h^\pi(s, \hat{b}) > -\infty$, then $\hat{C}_h^\pi(s, \hat{b}) \leq \hat{b} + \ell(S+1)(H - h + 1)$.*

**Theorem 5.4** (Bicriteria). *For any SR-criterion CMDP with polynomially-bounded costs and $\epsilon > 0$, the choice of $\ell \stackrel{\text{def}}{=} \frac{\epsilon}{1+(S+1)H}$ ensures Algorithm 4 is a $(0, \epsilon)$-bicriteria running in polynomial time $O\left(H^{6m+1} S^{4m+2} A \|c_{max} - c_{min}\|_\infty^{3m} / \epsilon^{3m}\right)$.*

**Corollary 5.5** (Relative). *For any $\epsilon > 0$, the choice of $\ell \stackrel{\text{def}}{=} \frac{\epsilon}{B(H(S+1)+1)}$ ensures Algorithm 4 is a polynomial time $(0, 1 + \epsilon)$-relative bicriteria for the class of polynomial-budget-bounded-cost CMDPs with SR-cost criteria. This includes all SR-criterion CMDPs with non-negative costs.*

*Remark* 5.6 (Chance Constraints). Technically, for chance constraints, we first create a cost-augmented MDP that is initially passed into the input. This allows us to write chance constraints in the SR form. Consequently, the $S$ term in Theorem 5.4 is really a larger augmented $S$. To achieve $\epsilon$ cost violation, (McMahan and Zhu, 2024b) showed that an augmented space of size $O(SH^2 \|c_{max} - c_{min}\|_\infty / \epsilon)$ is needed, which still results in a polynomial-time complexity.

*Remark* 5.7 (Approximation Optimality). (McMahan and Zhu, 2024b) showed that our assumptions on cost bounds are necessary to achieve polynomial-time approximations. Thus, our approximation guarantees are the best possible. Moreover, we can show that our dependency on the number of constraints is also unavoidable. This is formalized in Proposition 5.8.

**Proposition 5.8** (Multi-Constraint Hardness). *If $m = \Omega(n^{1/d})$ for some constant $d$, then computing an $\epsilon$-feasible policy for a CMDP is NP-hard for any $\epsilon > 0$.*

### 5.1. Continuous MDPs

We also show that approximations are possible in infinite state settings under certain continuity assumptions.

**Assumption 5.9** (Continuity). We assume the caMDP $M$ is Lipschitz continuous. Formally, we require that (1) $S = [s_{min}, s_{max}]$, (2) the reward function is $\lambda_r$ Lipschitz, (3) the cost function is $\lambda_c$ Lipschitz, (4) the transitions are $\lambda_p$ Lipschitz – each with respect to the state input, and (5) each of these quantities is polynomial-sized in the input representation. For SR-criterion CMDPs, we also assume that

$f$ has a natural finite equivalent denoted $\tilde{f}$, $g$ is a sublinear short map, and $f_{s'} z \leq (s_{max} - s_{min})$ for any constant $z$.

All we need to do is discretize the state space, and run our previous algorithm on the following discretized CMDP.

**Definition 5.10** (Discretized CMDP). Given any SR-criterion CMDP $(M, C, B)$, we define the *discretized CMDP* $(\tilde{M}, \tilde{C}, B)$ where $\tilde{M} = (H, \tilde{\mathcal{S}}, \mathcal{A}, \tilde{P}, R, C, \tilde{s}_0)$ is the discretized caMDP defined by,

1. $\tilde{\mathcal{S}}_h \stackrel{\text{def}}{=} \{\lceil s \rceil_\ell \mid s \in \mathcal{S}\}$

2. $\tilde{P}_h(\tilde{s}' \mid \tilde{s}, a) \stackrel{\text{def}}{=} \int_{s'=\tilde{s}'}^{\tilde{s}'+\ell} P_h(s' \mid \tilde{s}, a) ds'$

3. $\tilde{s}_0 \stackrel{\text{def}}{=} (\lceil s_0 \rceil_\ell, B)$

and $\tilde{C}$ is the cost criterion defined by replacing $f_{s'}$ with its natural finite equivalent $\tilde{f}$.

We see that discretization results in a small impact to both the value and cost that depend on the continuity parameters.

**Lemma 5.11** (Discretization). *For all $h \in [H+1]$, $\tau_h \in \mathcal{H}_h$, and $\pi \in \Pi^D$, we let $\tilde{\tau}_h$ denote $\tau_h$ with each state $s_t$ rounded to $\lceil s_t \rceil_\ell$. Then, we have that $\tilde{V}_h^\pi(\tilde{\tau}_h) \geq V_h^*(\tau_h) - \ell(\lambda_r + \lambda_p) H r_{max}(s_{max} - s_{min})(H - h + 1)$ and $\tilde{C}_h^\pi(\tilde{\tau}_h) \leq C_h^*(\tau_h) + \ell(\lambda_c + \lambda_p) H c_{max}(s_{max} - s_{min})(H - h + 1)$. For almost-sure/anytime constraints, the cost incurs an additional factor of $1/\tilde{p}_{min}$, where $\tilde{p}_{min}$ denotes the smallest non-zero transition probability for $\tilde{M}$.*

Overall, using our previous bicriteria on $\tilde{M}$ yields our approximation results.

**Theorem 5.12** (Continuous Bicriteria). *For any SR-criterion CMDP satisfying Assumption 5.9 and any $\epsilon > 0$, the choice of discretization $\ell_d \stackrel{\text{def}}{=} \frac{\epsilon/2}{(\lambda_r + \lambda_c + \lambda_p) H \max(c_{max}, r_{max})(s_{max} - s_{min})}$ and approximation $\ell_a \stackrel{\text{def}}{=} \frac{\epsilon/2}{1+(S+1)H}$ ensures Algorithm 4($\tilde{M}$) is a $(\epsilon, \epsilon)$-bicriteria running in time $O\left(H^{6m+1} \tilde{S}^{4m+2} A \|c_{max} - c_{min}\|_\infty^{3m} / \epsilon^{3m}\right)$, where $\tilde{S} = O((\lambda_r + \lambda_c + \lambda_p) H \max(c_{max}, r_{max})(s_{max} - s_{min})^2 / \epsilon)$. This time is polynomial so long as $|s_{max} - s_{min}| = O(|M|)$. Moreover, almost-sure/anytime constraints enjoy the same guarantee with an additional factor of $\tilde{p}_{min}$ in $\tilde{S}$.*

**Corollary 5.13** (Simplified). *For continuous-state SR-criterion CMDPs satisfying Assumption 5.9, there exist polynomial-time $(\epsilon, \epsilon)$-bicriteria solutions for expectation constraints, almost-sure constraints, anytime-almost-sure constraints, and any combinations of these constraints.*

## 6. Conclusion

In this work, we studied the question of whether polynomial-time approximation algorithms exist for many of the clas-

sic formulations studied in the CRL literature. We conclude that for the vast majority of constraints, including all the standard constraints, polynomial-time approximability is possible. We demonstrated this phenomenon by developing polynomial-time bicriteria approximations with the strongest possible guarantees for a general class of constraints that can be written in a form that satisfies general policy evaluation equations. Overall, our work resolves the polynomial-time approximability of many settings, some of which have lacked any polynomial-time algorithm for over a decade. In particular, we are the first to develop a polynomial-time algorithm with any kind of guarantee for chance constraints and non-homogeneous constraints.

## Impact Statement

This paper presents work whose goal is to advance the field of Machine Learning. There are many potential societal consequences of our work, none which we feel must be specifically highlighted here.

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

# A. Proofs for Section 2

## A.1. Proof of Proposition 2.3

*Proof.*

**Expectation Constraints.** We define $C_M^\pi \stackrel{\text{def}}{=} \mathbb{E}_M\left[\sum_{h=1}^H c_H\right]$. Under this definition, the standard policy evaluation equations imply that,

$$C_h^\pi(\tau_h) = c_h(s, a) + \sum_{s'} P_h(s' \mid s, a) C_{h+1}^\pi(\tau_{h+1}). \tag{8}$$

It is then clear that this can be written in $(f, g)$-form for $f$ being summation and $g$ being the identity. It is easy to see that these functions have the desired properties.

**Chance Constraints.** Let $M^0$ denote the initial caMDP. We define $C_{M^0}^\pi \stackrel{\text{def}}{=} \mathbb{P}_M^\pi\left[\sum_{h=1}^H c_h > B\right]$. We see that the probability can be recursively decomposed as follows for the anytime variant:

$$C_h^\pi(\tau_h, \bar{c}) = [c_h(s, a) + \bar{c} > B] + \sum_{s'} P_h(s' \mid s, a) C_{h+1}^\pi(\tau_{h+1}, c_h(s, a) + \bar{c}). \tag{9}$$

For the general invariant, we only include the indicator term at step $H$. To write this into the desired form, we can define a cost-augmented MDP $M$ that keeps track of the cumulative cost at each step as in (McMahan and Zhu, 2024a). In particular, the anytime variant has the immediate cost defined to be $c_h((s, \bar{c}), a) \stackrel{\text{def}}{=} [c_h(s, a) + \bar{c} > B]$. Then, it is clear that the expected cost for the new $M$ exactly corresponds to the probability cost. Thus, the claim holds.

**Almost-sure Constraints.** We define $C_M^\pi \stackrel{\text{def}}{=} \max_{\substack{\tau_{H+1} \\ \mathbb{P}_M^\pi[\tau_{H+1}]>0}} \left[\sum_{h=1}^H c_H\right]$ to be the worst case cost. Under this definition, it is known that the worst-case cost decomposes into,

$$C_h^\pi(\tau_h) = c_h(s, a) + \max_{s'}[P_h(s' \mid s, a) > 0] C_{h+1}^\pi(\tau_{h+1}). \tag{10}$$

It is then clear that this can be written in $(f, g)$-form for $f$ being maximum and $g$ being the indicator. Properties of maximum imply that $\max_{s'}(C(s') + \epsilon) \leq \max_{s'} C(s') + \epsilon$. Thus, the total combination is a short map, and the rest of the needed properties can be seen to hold. The anytime variant follows similarly.

$\square$

## A.2. Proof of Theorem 2.5

*Proof.* The theorem follows immediately by translating the results on the SR-criterion into their original forms in the proof above. $\square$

# B. Proof for Section 3

## B.1. Helpful Technical Lemmas

**Definition B.1** (Budget Space). For any $s \in \mathcal{S}$, we define $\mathcal{B}_{H+1}(s) \stackrel{\text{def}}{=} \{0\}$, and for any $h \in [H]$,

$$\mathcal{B}_h(s) \stackrel{\text{def}}{=} \bigcup_a \bigcup_{\mathbf{b} \in \times_{s'} \mathcal{B}_{h+1}(s')} \left\{ c_h(s, a) + \underset{s'}{f} \, g(P_h(s' \mid s, a), b_{s'}) \right\}. \tag{11}$$

We define $\mathcal{B} \stackrel{\text{def}}{=} \bigcup_{h,s} \mathcal{B}_h(s)$.

**Lemma B.2** (Budget Space Intuition). *For all $s \in \mathcal{S}$ and $h \in [H+1]$,*

$$\mathcal{B}_h(s) = \left\{ b \in \mathbb{R}^d \mid \exists \pi \in \Pi^D, \tau_h \in \mathcal{H}_h, \, (s = s_h(\tau_h) \wedge C_h^\pi(\tau_h) = b) \right\}, \tag{12}$$

*and $|\mathcal{B}_h(s)| \leq A^{\sum_{t=h}^H S^{H-t}}$. Thus, $\mathcal{B}$ can be computed in finite time using backward induction.*

*Proof.* We proceed by induction on $h$. Let $s \in \mathcal{S}$ be arbitrary.

**Base Case.** For the base case, we consider $h = H + 1$. In this case, we know that for any $\pi \in \Pi^D$ and any $\tau \in \mathcal{H}_{H+1}$, $C_{H+1}^\pi(\tau_{H+1}) = 0 \in \{0\} = \mathcal{B}_{H+1}(s)$ by definition. Furthermore, $|\mathcal{B}_{H+1}(s)| = 1 = A^0 = A^{\sum_{t=H+1}^H S^t}$.

**Inductive Step.** For the inductive step, we consider $h \leq H$. In this case, we know that for any $\pi \in \Pi^D$ and any $\tau_h \in \mathcal{H}_h$, if $s = s_h(\tau_h)$ and $a = \pi_h(\tau_h)$, then the policy evaluation equations imply,

$$C_h^\pi(\tau_h) = c_h(s,a) + \underset{s'}{f} \, g(P_h(s' \mid s,a), C_{h+1}^\pi(\tau_h, a, s')).$$

We know by the induction hypothesis that $V_{h+1}^\pi(\tau_h, a, s') \in \mathcal{B}_{h+1}(s')$. Thus, by (11), $C_h^\pi(\tau_h) \in \mathcal{B}_h(s)$. Lastly, we see by (11) and the induction hypothesis that,

$$|\mathcal{B}_h(s)| \leq A \prod_{s'} |\mathcal{B}_{h+1}(s')| \leq A \prod_{s'} A^{\sum_{t=h+1}^H S^{H-t}} = A^{1+S\sum_{t=h+1}^H S^{H-t}} = A^{\sum_{t=h}^H S^{H-t}}.$$

This completes the proof. $\qquad\square$

## B.2. Proof of Lemma 3.2

*Proof.* First, let $V_h^*(\tau_h, b)$ denote the supremum in (5). We proceed by induction on $h$.

**Base Case.** For the base case, we consider $h = H + 1$. Definition 2.1 implies that $C_{H+1}^\pi(\tau_{H+1}) = 0$ for any $\pi \in \Pi^D$. Thus, there exists a $\pi \in \Pi^D$ satisfying $C_{H+1}^\pi(\tau_{H+1}) \leq b$ if and only if $b \geq 0$. We also know by definition that any policy $\pi$ satisfies $V_{H+1}^\pi(\tau_{H+1}) = 0$ and if no feasible policy exists $V_{H+1}^*(\tau_{H+1}, b) = -\infty$ by convention. Therefore, we see that $V_{H+1}^*(\tau_{H+1}, b) = -\chi_{\{b \geq 0\}}$. Then, by definition of $\bar{V}_{H+1}^*$, it follows that,

$$\bar{V}_{H+1}^*(s, b) = -\chi_{\{b \geq 0\}} = V_{H+1}^*(\tau_{H+1}, b).$$

**Inductive Step.** For the inductive step, we consider any $h \leq H$. If $V_h^*(\tau_h, b) = -\infty$, then trivially $\bar{V}_h^*(s, b) \geq V_h^*(\tau_h, b)$. Instead, suppose that $V_h^*(\tau_h, b) > -\infty$. Then, there must exist a $\pi \in \Pi^D$ satisfying $C_h^\pi(\tau_h) \leq b$. Let $a^* = \pi_h(\tau_h)$. By (SR), we know that,

$$C_h^\pi(\tau_h) = c_h(s, a^*) + \underset{s'}{f} \, g\left(P_h(s' \mid s, a^*)\right) C_{h+1}^\pi(\tau_h, a^*, s').$$

For each $s' \in \mathcal{S}$, define $b_{s'}^* \overset{\text{def}}{=} C_{h+1}^\pi(\tau_h, a^*, s')$ and observe that $b_{s'}^* \in \mathcal{B}$. Thus, we see that $(a^*, \mathbf{b}^*) \in \mathcal{A} \times \mathcal{B}^S$ and $c_h(s, a) + f_{s'} \, g(P_h(s' \mid s, a)) b_{s'} \leq b$, so $(a^*, \mathbf{b}^*) \in \bar{\mathcal{A}}_h(s, b)$ by definition.

Since $\pi$ satisfies $C_{h+1}^\pi(\tau_h, a^*, s') \leq b_{s'}^*$, we see that $V_{h+1}^*(s', b_{s'}^*) \geq V_{h+1}^\pi(\tau_h, a^*, s')$. Thus, the induction hypothesis implies $\bar{V}_{h+1}^*(s', b_{s'}^*) \geq V_{h+1}^*(s', b_{s'}^*) \geq V_{h+1}^\pi(\tau_h, a^*, s')$. The optimality equations for $\bar{M}$ then give us,

$$
\begin{aligned}
\bar{V}_h^*(s, b) &= \max_{\bar{a} \in \bar{\mathcal{A}}_h(s,b)} \bar{r}_h((s,b), \bar{a}) + \sum_{\bar{s}'} \bar{P}_h(\bar{s}' \mid (s,b), \bar{a}) \bar{V}_{h+1}^*(\bar{s}') \\
&= \max_{(a,\mathbf{b}) \in \bar{\mathcal{A}}_h(s,b)} r_h(s, a) + \sum_{s'} P_h(s' \mid s, a) \bar{V}_{h+1}^*(s', b_{s'}) \\
&\geq r_h(s, a^*) + \sum_{s'} P_h(s' \mid s, a^*) \bar{V}_{h+1}^*(s', b_{s'}^*) \\
&\geq r_h(s, a^*) + \sum_{s'} P_h(s' \mid s, a^*) V_{h+1}^\pi(\tau_h, a, s') \\
&= V_h^\pi(\tau_h).
\end{aligned}
$$

The second line used the definition of each quantity in $\bar{M}$. The first inequality used the fact that $(a^*, \mathbf{b}^*) \in \bar{\mathcal{A}}_h(s, b)$. The second inequality used the induction hypothesis. The final equality used the deterministic policy evaluation equations.

Since $\pi$ was an arbitrary feasible policy for the optimization defining $V_h^*(\tau_h, b)$, we see that $\bar{V}_h^*(s, b) \geq V_h^*(\tau_h, b)$. This completes the proof.

$\qquad\square$

**B.3. Proof of Lemma 3.3**

*Proof.* We proceed by induction on $h$.

**Base Case.** For the base case, we consider $h = H + 1$. By definition and assumption, $\bar{V}_{H+1}^\pi(s, b) = -\chi_{\{b \geq 0\}} > -\infty$. Thus, it must be the case that $b \geq 0$ and so by Definition 2.1 $\bar{C}_{H+1}^\pi(s, b) = 0 \leq b$.

**Inductive Step.** For the inductive step, we consider any $h \leq H$. We decompose $\pi_h(s, b) = (a, \mathbf{b})$ where we know $(a, \mathbf{b}) \in \bar{\mathcal{A}}_h(s, b)$ since $\bar{V}_h^\pi(s, b) > -\infty$ [2]. Moreover, it must be the case that for any $s' \in \mathcal{S}$ with $P_h(s' \mid s, a) > 0$ that $\bar{V}_{h+1}^\pi(s', b_{s'}) > -\infty$ otherwise the average reward would be $-\infty$ which would imply a contradiction:

$$\bar{V}_h^\pi(s, b) = r_h(s, a) + \sum_{s'} P_h(s' \mid s, a) \bar{V}_{h+1}^\pi(s', b_{s'})$$
$$= r_h(s, a) + \ldots + P_h(s' \mid s, a)(-\infty) + \ldots$$
$$= -\infty.$$

Thus, the induction hypothesis implies that $\bar{C}_{h+1}^\pi(s', b_{s'}) \leq b_{s'}$ for any such $s' \in \mathcal{S}$. By (SR), we see that,

$$\bar{C}_h^\pi(s, b) = c_h(s, a) + \underset{s'}{f} \, g(P_h(s' \mid s, a)) \bar{C}_{h+1}^\pi(s', b_{s'})$$
$$\leq c_h(s, a) + \underset{s'}{f} \, g(P_h(s' \mid s, a)) b_{s'}$$
$$\leq b.$$

The second line used the fact that $f$ is non-decreasing and $g$ is a non-negative scalar. The third line used the fact that $(a, \mathbf{b}) \in \bar{\mathcal{A}}_h(s, b)$. This completes the proof. $\square$

**B.4. Proof of Theorem 3.4**

*Proof.* If $\bar{V}_1^*(s_0, B) = -\infty$, then we know by Lemma 3.2 that,

$$-\infty = \bar{V}_1^*(s_0, B) \geq \sup_{\pi \in \Pi^D} V_1^\pi(s_0)$$
$$\text{s.t. } C_1^\pi(s_0) \leq B. \tag{13}$$

In other words, no feasible $\pi$ exists, so Algorithm 1 reporting "Infeasible" is correct. On the other hand, suppose that $\bar{V}_1^*(s_0, B) > -\infty$ and let $\pi^*$ be any solution to the optimality equations for $\bar{M}$. By Lemma 3.3, we know that $C_1^\pi(s_0, B) \leq B$ implying that $\pi^*$ is a feasible solution. Moreover, Lemma 3.2 again tells us that,

$$\bar{V}_1^{\pi^*}(s_0, B) = \bar{V}_1^*(s_0, B) \geq \sup_{\pi \in \Pi^D} V_1^\pi(s_0)$$
$$\text{s.t. } C_1^\pi(s_0) \leq B. \tag{14}$$

Thus, $\pi^*$ is an optimal solution to (CON) and Algorithm 1 correctly returns it. Therefore, in all cases, Algorithm 1 is correct. $\square$

## C. Proofs for Section 4

Formally, $\hat{f}_{h,\mathbf{b}}^{s,a}$ can be defined recursively by $\hat{f}_{h,\mathbf{b}}^{s,a}(t, \hat{F}) \stackrel{\text{def}}{=} \hat{f}_{h,\mathbf{b}}^{s,a}\left(t + 1, \left[f(\hat{F}, g(P_h(t \mid s, a)) b_t)\right]_\ell\right)$ with base case $\hat{f}_{h,\mathbf{b}}^{s,a}(S + 1, \hat{F}) \stackrel{\text{def}}{=} \hat{F}$.

---

[2] By convention, we assume $\max \varnothing = -\infty$

### C.1. Proof of Lemma 4.2

*Proof.* First, we show that,

$$\bar{V}^{s,a}_{h,b}(t, \hat{F}) = \max_{\mathbf{b} \in \mathcal{B}^{S-t+1}} \sum_{s'=t}^{S} P_h(s' \mid s, a)\bar{V}^*_{h+1}(s', b_{s'}) \tag{15}$$

$$\text{s.t.} \quad c_h(s, a) + f^{s,a}_{h,\mathbf{b}}(t, F) \leq b,$$

For notational simplicity, we define $\bar{V}^{s,a}_{h,\mathbf{b}}(t) \stackrel{\text{def}}{=} \sum_{s'=t}^{S} P_h(s' \mid s, a)\bar{V}^*_{h+1}(s', b_{s'})$. We proceed by induction on $t$.

**Base Case.** For the base case, we consider $t = S + 1$. By definition, we know that $\bar{V}^{s,a}_{h,b}(t, F) = -\chi_{\{c_h(s,a)+F \leq b\}}$. We just need to show that the maximum in (15) also matches this expression. First, observe objective is the empty summation, which is 0. Also, $f^{s,a}_{h,\mathbf{b}}(S + 1, F) = F$, so the constraint is satisfied iff $c_h(s, a) + F \leq b$. Thus, the maximum is 0 when $c_h(s, a) + F \leq b$ and is $-\infty$ due to infeasibility otherwise. In other words, it equals $-\chi_{\{c_h(s,a)+F \leq b\}}$ as was to be shown.

**Inductive Step.** For the inductive step, we consider any $t \leq S$. From (6), we see that,

$$\bar{V}^{s,a}_{h,b}(t, F) = \max_{b_t \in \mathcal{B}} P_h(t \mid s, a)\bar{V}^*_{h+1}(t, b_t) + \bar{V}^{s,a}_{h,b}(t + 1, f(F, g(P_h(t \mid s, a))b_t))$$

$$= \max_{b_t \in \mathcal{B}} P_h(t \mid s, a)\bar{V}^*_{h+1}(t, b_t) + \max_{\substack{\mathbf{b} \in \mathcal{B}^{S-t}, \\ c_h(s,a)+f^{s,a}_{h,\mathbf{b}}(t+1,f(F,g(P_h(t\mid s,a))b_t)) \leq b}} \bar{V}^{s,a}_{h,\mathbf{b}}(t + 1)$$

$$= \max_{b_t \in \mathcal{B}} \max_{\substack{\mathbf{b} \in \mathcal{B}^{S-t}, \\ c_h(s,a)+f^{s,a}_{h,\mathbf{b}}(t+1,f(F,g(P_h(t\mid s,a))b_t)) \leq b}} P_h(t \mid s, a)\bar{V}^*_{h+1}(t, b_t) + \bar{V}^{s,a}_{h,\mathbf{b}}(t + 1)$$

$$= \max_{\substack{\mathbf{b} \in \mathcal{B}^{S-t+1}, \\ c_h(s,a)+f^{s,a}_{h,\mathbf{b}}(t+1,f(F,g(P_h(t\mid s,a))b_t)) \leq b}} P_h(t \mid s, a)\bar{V}^*_{h+1}(t, b_t) + \bar{V}^{s,a}_{h,\mathbf{b}}(t + 1)$$

$$= \max_{\substack{\mathbf{b} \in \mathcal{B}^{S-t+1}, \\ c_h(s,a)+f^{s,a}_{h,\mathbf{b}}(t,F) \leq b}} P_h(t \mid s, a)\bar{V}^*_{h+1}(t, b_t) + \bar{V}^{s,a}_{h,\mathbf{b}}(t + 1)$$

$$= \max_{\substack{\mathbf{b} \in \mathcal{B}^{S-t+1}, \\ c_h(s,a)+f^{s,a}_{h,\mathbf{b}}(t,F) \leq b}} \bar{V}^{s,a}_{h,\mathbf{b}}(t)$$

The second line used the induction hypothesis. The third line used the fact that the first term is independent of future $b$ values. The fourth line used the properties of maximum. The fourth line used the recursive definition of $f^{s,a}_{h,\mathbf{b}}(t, F)$. The last line used the recursive definition of $\bar{V}^{s,a}_{h,\mathbf{b}}(t)$.

For the second claim, we observe that,

$$\bar{V}^*_h(s, b) = \max_{\substack{a,\mathbf{b}, \\ c_h(s,a)+f_{s'} g(P_h(s'\mid s,a))b_{s'} \leq b}} r_h(s, a) + \sum_{s'} P_h(s' \mid s, a)\bar{V}^*_{h+1}(s', b_{s'})$$

$$= \max_{\substack{a,\mathbf{b}, \\ c_h(s,a)+f^{s,a}_{h,\mathbf{b}}(1,0) \leq b}} r_h(s, a) + \sum_{s'} P_h(s' \mid s, a)\bar{V}^*_{h+1}(s', b_{s'})$$

$$= \max_a \max_{\substack{\mathbf{b}, \\ c_h(s,a)+f^{s,a}_{h,\mathbf{b}}(1,0) \leq b}} r_h(s, a) + \sum_{s'} P_h(s' \mid s, a)\bar{V}^*_{h+1}(s', b_{s'})$$

$$= \max_a r_h(s, a) + \max_{\substack{\mathbf{b}, \\ c_h(s,a)+f^{s,a}_{h,\mathbf{b}}(1,0) \leq b}} \sum_{s'} P_h(s' \mid s, a)\bar{V}^*_{h+1}(s', b_{s'})$$

$$= \max_a r_h(s, a) + \bar{V}^{s,a}_{h,b}(1, 0).$$

$\square$

## C.2. Proof of Lemma 4.5

*Proof.* Recall, as in the proof of Lemma 4.2, we define $\bar{V}_{h,\mathbf{b}}^{s,a}(t) \stackrel{\text{def}}{=} \sum_{s'=t}^{S} P_h(s' \mid s,a)\bar{V}_{h+1}^{*}(s', b_{s'})$ to simplify expressions. We proceed by induction on $t$.

**Base Case.** For the base case, we consider $t = S+1$. By definition, we know that $\hat{V}_{h,b}^{s,a}(t, \hat{F}) = -\chi_{\{c_h(s,a)+\hat{F} \leq \kappa(b)\}}$. We just need to show that the maximum in (7) also matches this expression. First, observe objective is the empty summation, which is 0. Also, $\hat{f}_{h,\mathbf{b}}^{s,a}(S+1, F) = F$, so the constraint is satisfied iff $c_h(s,a) + \hat{F} \leq \kappa(b)$. Thus, the maximum is 0 when $c_h(s,a) + \hat{F} \leq \kappa(b)$ and is $-\infty$ due to infeasibility otherwise. In other words, it equals $-\chi_{\{c_h(s,a)+\hat{F} \leq \kappa(b)\}}$ as was to be shown.

**Inductive Step.** For the inductive step, we consider any $t \leq S$. From (ADP), we see that,

$$\hat{V}_{h,b}^{s,a}(t, \hat{F}) = \max_{b_t \in \mathcal{B}} P_h(t \mid s,a)\bar{V}_{h+1}^{*}(t, b_t) + \hat{V}_{h,b}^{s,a}\left(t+1, \left\lceil f\left(\hat{F}, g(P_h(t \mid s,a))b_t\right)\right\rceil_\ell\right)$$

$$= \max_{b_t \in \mathcal{B}} P_h(t \mid s,a)\bar{V}_{h+1}^{*}(t, b_t) + \max_{\substack{\mathbf{b} \in \mathcal{B}^{S-t}, \\ c_h(s,a)+\hat{f}_{h,\mathbf{b}}^{s,a}\left(t+1, \left\lceil f\left(\hat{F}, g(P_h(t \mid s,a))b_t\right)\right\rceil_\ell\right) \leq \kappa(b)}} \bar{V}_{h,\mathbf{b}}^{s,a}(t+1)$$

$$= \max_{b_t \in \mathcal{B}} \max_{\substack{\mathbf{b} \in \mathcal{B}^{S-t}, \\ c_h(s,a)+\hat{f}_{h,\mathbf{b}}^{s,a}\left(t+1, \left\lceil f\left(\hat{F}, g(P_h(t \mid s,a))b_t\right)\right\rceil_\ell\right) \leq \kappa(b)}} P_h(t \mid s,a)\bar{V}_{h+1}^{*}(t, b_t) + \bar{V}_{h,\mathbf{b}}^{s,a}(t+1)$$

$$= \max_{\substack{\mathbf{b} \in \mathcal{B}^{S-t+1}, \\ c_h(s,a)+\hat{f}_{h,\mathbf{b}}^{s,a}\left(t+1, \left\lceil f\left(\hat{F}, g(P_h(t \mid s,a))b_t\right)\right\rceil_\ell\right) \leq \kappa(b)}} P_h(t \mid s,a)\bar{V}_{h+1}^{*}(t, b_t) + \bar{V}_{h,\mathbf{b}}^{s,a}(t+1)$$

$$= \max_{\substack{\mathbf{b} \in \mathcal{B}^{S-t+1}, \\ c_h(s,a)+\hat{f}_{h,\mathbf{b}}^{s,a}\left(t,\hat{F}\right) \leq \kappa(b)}} P_h(t \mid s,a)\bar{V}_{h+1}^{*}(t, b_t) + \bar{V}_{h,\mathbf{b}}^{s,a}(t+1)$$

$$= \max_{\substack{\mathbf{b} \in \mathcal{B}^{S-t+1}, \\ c_h(s,a)+\hat{f}_{h,\mathbf{b}}^{s,a}\left(t,\hat{F}\right) \leq \kappa(b)}} \bar{V}_{h,\mathbf{b}}^{s,a}(t)$$

The second line used the induction hypothesis. The third line used the fact that the first term is independent of future $b$ values. The fourth line used the properties of maximum. The fourth line used the recursive definition of $\hat{f}_{h,\mathbf{b}}^{s,a}(t, \hat{F})$. The last line used the recursive definition of $\bar{V}_{h,\mathbf{b}}^{s,a}(t)$.

For the second claim, we simply observe without rounding that (ADP) is the same as (6). Thus, Lemma 4.2 yields the result. $\qquad\square$

## C.3. Proof of Theorem 4.7

*Proof.* The fact that Algorithm 3 correctly solves any $\bar{M}$ follows from the fact that (AU) is equivalent to (BU) via Lemma 4.5.

For the time complexity claim, observe that the number of subproblems considered is $O(HS^2A|\mathcal{B}||\hat{\mathcal{F}}|)$ and the time needed per subproblem is $O(|\mathcal{B}|)$ to explicitly optimize each artificial budget. Thus, the running time is $O(HS^2A|\mathcal{B}|^2|\hat{\mathcal{F}}|)$. We can further analyze $|\hat{\mathcal{F}}|$ in terms of the original input variables. First, we claim that $\hat{\mathcal{F}} \subseteq [b_{min}, b_{max} + \ell S]$. To see this, observe that the rounded input at state $t+1$ is,

$$f(\hat{F}, \lceil b_t \rceil_\ell) \geq f(F, b_t) = \overset{t}{\underset{s'=1}{f}} g(P_h(s' \mid s,a))b_{s'} \geq \overset{t}{\underset{s'=1}{f}} g(P_h(s' \mid s,a))b_{min} \geq b_{min}.$$

Here, we used the fact that $f$ is non-decreasing and the weighted combination is a short map rooted at 0. Similarly, we see,

$$f(\hat{F}, \lceil b_t \rceil_\ell) \leq f(F, \lceil b_t \rceil_\ell) + \ell(t-1)$$

$$\leq \overset{t}{\underset{s'=1}{f}} g(P_h(s' \mid s,a))(b_{s'} + \ell) + \ell(t-1)$$

$$\leq \overset{t}{\underset{s'=1}{f}} g(P_h(s' \mid s,a))b_{max} + \ell t$$

$$\leq b_{max} + \ell t.$$

Under this assumption, it is clear that the number of integer multiples of $\ell$ residing in this superset is $O((b_{max} + \ell S - b_{min})/\ell)$ per constraint. When considering all constraints at once, this becomes $O(\|b_{max} + \ell S - b_{min}\|_\infty^m / \ell^m) = O(\|b_{max} - b_{min}\|_\infty^m / \ell^m + S^m)$. Incorporating this bound into the runtime then gives $O(HS^{m+2}A|\mathcal{B}|^2 \|b_{max} - b_{min}\|_\infty^m / \ell^m)$.

Similar to the reasoning above, we can see the cost of any policy, and thus the artificial budget set, is contained within $[Hc_{min}, Hc_{max}]$. Using this fact, we get the final running time $O(H^{m+1}S^{m+2}A|\mathcal{B}|^2 \|c_{max} - c_{min}\|_\infty^m / \ell^m)$.

$\square$

# D. Proofs for Section 5

## D.1. Time-Space Error Lemmas

**Lemma D.1** (Time Error). *For any $h \in [H]$, $a \in \mathcal{A}$, if $\mathbf{b}' \leq \mathbf{b} + x$, then,*

$$f_{h,\mathbf{b}}^{s,a}(1,0) \leq f_{h,\mathbf{b}'}^{s,a}(1,0) \leq f_{h,\mathbf{b}}^{s,a}(1,0) + x. \tag{16}$$

*Here, we translate a scalar $x > 0$ into the vector $(x, \ldots, x)$.*

*Proof.* By definition of $f_{h,\mathbf{b}'}^{s,a}$,

$$
\begin{aligned}
f_{h,\mathbf{b}'}^{s,a}(1,0) &= f(0, \underset{s'}{f} g(P_h(s' \mid s, a))b'_{s'}) \\
&= \underset{s'}{f} g(P_h(s' \mid s, a))b'_{s'} \\
&\geq \underset{s'}{f} g(P_h(s' \mid s, a))b_{s'} \\
&= f(0, \underset{s'}{f} g(P_h(s' \mid s, a))b_{s'}) \\
&= f_{h,\mathbf{b}}^{s,a}(1,0).
\end{aligned}
$$

The second and fourth lines used the fact that $f$ is identity preserving. The inequality uses the fact that $f$ is non-decreasing and $g$ is a non-negative scalar, so the total weighted combination is also non-decreasing.

Similarly, we see that,

$$
\begin{aligned}
f_{h,\mathbf{b}'}^{s,a}(1,0) &= f(0, \underset{s'}{f} g(P_h(s' \mid s, a))b'_{s'}) \\
&= \underset{s'}{f} g(P_h(s' \mid s, a))b'_{s'} \\
&\leq \underset{s'}{f} g(P_h(s' \mid s, a))(b_{s'} + x) \\
&\leq \underset{s'}{f} g(P_h(s' \mid s, a))b_{s'} + x \\
&= f(0, \underset{s'}{f} g(P_h(s' \mid s, a))b_{s'}) + x \\
&= f_{h,\mathbf{b}}^{s,a}(1,0) + x.
\end{aligned}
$$

The second and fifth lines used the fact that $f$ is identity preserving. The first inequality again uses the fact that the weighted combination is non-decreasing. The second inequality follows since the weighted combination is a short map with respect to the infinity norm.

In particular, since $|\alpha(y) - \alpha(z)| \leq \|y - z\|_\infty$ holds for any infinity-norm short map $\alpha$, we see that $|\alpha(y+z) - \alpha(y)| \leq \|z\|_\infty$. Moreover, if $\alpha$ is non-decreasing and $z$ is a positive scaler treated as a vector, we further have $\alpha(y + z) - \alpha(y) = |\alpha(y + z) - \alpha(y)| \leq \|z\|_\infty = z$. This final inequality immediately implies that $\alpha(y + z) \leq \alpha(y) + z$. When $\alpha$ is vector-valued, this inequality holds component-wise. $\square$

Since $f$ is associative, we can define $f_{h,\mathbf{b}}^{s,a}(t, F) = f(F, \underset{s'=t}{f^S} g(P_h(s' \mid s, a))b_{s'})$ either forward recursively or backward recursively.

**Lemma D.2** (Space Error). *For any $h \in [H]$, $a \in \mathcal{A}$, $\mathbf{b} \in \mathbb{R}^{m \times S}$, $u \in \mathbb{R}^m$, and $t \in [S+1]$,*

$$f_{h,\mathbf{b}}^{s,a}(t,u) \leq \hat{f}_{h,\mathbf{b}}^{s,a}(t,u) \leq f_{h,\mathbf{b}}^{s,a}(t,u) + (S-t+1)\ell. \tag{17}$$

*Proof.* We proceed by induction on $t$.

**Base Case.** For the base case, we consider $t = S+1$. By definition, we have that $\hat{f}_{h,\mathbf{b}}^{s,a}(S+1,u) = u = f_{h,\mathbf{b}}^{s,a}(S+1,u)$. Thus, the claim holds.

**Inductive Step.** For the inductive step, we consider any $t \leq S$. The recursive definition of $\hat{f}_{h,\mathbf{b}}^{s,a}$ implies,

$$
\begin{aligned}
\hat{f}_{h,\mathbf{b}}^{s,a}(t,u) &= \hat{f}_{h,\mathbf{b}}^{s,a}(t+1, \lceil f(u, g(P_h(t \mid s,a))b_t) \rceil_\ell) \\
&\geq f_{h,\mathbf{b}}^{s,a}(t+1, \lceil f(u, g(P_h(t \mid s,a))b_t) \rceil_\ell) \\
&= f\left( \lceil f(u, g(P_h(t \mid s,a))b_t) \rceil_\ell, \underset{s'=t+1}{\overset{S}{f}} g(P_h(s' \mid s,a)b_{s'}) \right) \\
&\geq f\left( f(u, g(P_h(t \mid s,a))b_t), \underset{s'=t+1}{\overset{S}{f}} g(P_h(s' \mid s,a)b_{s'}) \right) \\
&= f\left( u, f(g(P_h(t \mid s,a))b_t, \underset{s'=t+1}{\overset{S}{f}} g(P_h(s' \mid s,a)b_{s'})) \right) \\
&= f_{h,\mathbf{b}}^{s,a}(t,u).
\end{aligned}
$$

The first inequality used the induction hypothesis to replace $\hat{f}$ with $f$, and the second inequality used that $f$ is non-decreasing in either input and $\lceil b_t \rceil_\ell \geq b_t$. The other lines use $f$'s associativity.

Similarly, we see that,

$$
\begin{aligned}
\hat{f}_{h,\mathbf{b}}^{s,a}(t,u) &= \hat{f}_{h,\mathbf{b}}^{s,a}(t+1, \lceil f(u, g(P_h(t \mid s,a))b_t) \rceil_\ell) \\
&\leq f_{h,\mathbf{b}}^{s,a}(t+1, \lceil f(u, g(P_h(t \mid s,a))b_t) \rceil_\ell) + (S-t)\ell \\
&= f\left( \lceil f(u, g(P_h(t \mid s,a))b_t) \rceil_\ell, \underset{s'=t+1}{\overset{S}{f}} g(P_h(s' \mid s,a)b_{s'}) \right) + (S-t)\ell \\
&\leq f\left( f(u, g(P_h(t \mid s,a))b_t) + \ell, \underset{s'=t+1}{\overset{S}{f}} g(P_h(s' \mid s,a)b_{s'}) \right) + (S-t)\ell \\
&\leq f\left( f(u, g(P_h(t \mid s,a))b_t), \underset{s'=t+1}{\overset{S}{f}} g(P_h(s' \mid s,a)b_{s'}) \right) + (S-t+1)\ell \\
&= f\left( u, f(g(P_h(t \mid s,a))b_t, \underset{s'=t+1}{\overset{S}{f}} g(P_h(s' \mid s,a)b_{s'})) \right) + (S-t+1)\ell \\
&= f_{h,\mathbf{b}}^{s,a}(t,u) + (S-t+1)\ell.
\end{aligned}
$$

The first inequality used the induction hypothesis to replace $\hat{f}$ with $f$. The second inequality used that $f$ is non-decreasing in either input and $\lceil x \rceil_\ell \leq x + \ell$. The third inequality used that $f$ is a short map in the first input. The other lines use $f$'s associativity.

This completes the proof. $\square$

### D.2. Proof of Lemma 5.2

*Proof.* We proceed by induction on $h$.

**Base Case.** For the base case, we consider $h = H+1$. Since $\lceil b \rceil_\ell \geq b$, we immediately see,

$$\hat{V}_{H+1}^*(s, \lceil b \rceil_\ell) = -\chi_{\{\lceil b \rceil_\ell \geq 0\}} \geq -\chi_{\{b \geq 0\}} = \bar{V}_{H+1}^*(s,b). \tag{18}$$

**Inductive Step.** For the inductive step, we consider any $h \leq H$. If $\bar{V}_h^*(s, b) = -\infty$, then trivially $\hat{V}_h^*(s, \lceil b \rceil_\ell) \geq \bar{V}_h^*(s, b)$. Now, suppose that $\bar{V}_h^*(s, b) > -\infty$. Let $\pi$ be a solution to the optimality equations for $\bar{M}$. Consequently, we know that $\bar{V}_h^\pi(s, b) = \bar{V}_h^*(s, b) > -\infty$, which implies $(a^*, \mathbf{b}^*) = \pi_h(s, b) \in \bar{\mathcal{A}}_h(s, b)$. By definition of $\bar{\mathcal{A}}_h(s, b)$,

$$c_h(s, a^*) + f_{h, \mathbf{b}^*}^{s, a^*}(1, 0) = c_h(s, a^*) + \underset{s'}{f} \, g(P_h(s' \mid s, a^*))b_{s'}^* \leq b \leq \lceil b \rceil_\ell. \tag{19}$$

For each $s' \in \mathcal{S}$, define $\hat{b}_{s'}^* \stackrel{\text{def}}{=} \lceil b_{s'}^* \rceil_\ell$. We show $(a^*, \hat{\mathbf{b}}_{s'}^*) \in \hat{\mathcal{A}}_h(s, \lceil b \rceil_\ell)$ as follows:

$$
\begin{aligned}
c_h(s, a^*) + \hat{f}_{h, \hat{\mathbf{b}}^*}^{s, a}(1, 0) &\leq c_h(s, a^*) + f_{h, \hat{\mathbf{b}}^*}^{s, a}(1, 0) + \ell S \\
&\leq c_h(s, a^*) + f_{h, \mathbf{b}^*}^{s, a}(1, 0) + \ell(S + 1) \\
&\leq \lceil b \rceil_\ell + \ell(S + 1) \\
&= \kappa(\lceil b \rceil_\ell).
\end{aligned}
$$

The first inequality follows from Lemma D.2. The second inequality follows from Lemma D.1 with $\hat{\mathbf{b}}^* \leq \mathbf{b}^* + \ell$. The third inequality follows from (19). The equality follows by definition of $\kappa$. Thus, $(a^*, \hat{\mathbf{b}}_{s'}^*) \in \hat{\mathcal{A}}_h(s, \lceil b \rceil_\ell)$.

Since $b_{s'}^* \in \mathcal{B}$ by definition, the induction hypothesis implies that $\hat{V}_{h+1}^*(s', \hat{b}_{s'}^*) \geq \bar{V}_{h+1}^*(s', b_{s'}^*) = \bar{V}_{h+1}^\pi(s', b_{s'}^*)$. The optimality equations for $\hat{M}$ then imply that,

$$
\begin{aligned}
\hat{V}_h^*(s, \lceil b \rceil_\ell) &= \max_{(a, \hat{\mathbf{b}}) \in \hat{\mathcal{A}}_h(s, b)} r_h(s, a) + \sum_{s'} P_h(s' \mid s, a) \hat{V}_{h+1}^* \left( s', \hat{b}_{s'} \right) \\
&\geq r_h(s, a^*) + \sum_{s'} P_h(s' \mid s, a) \hat{V}_{h+1}^* \left( s', \hat{b}_{s'}^* \right) \\
&\geq r_h(s, a^*) + \sum_{s'} P_h(s' \mid s, a) \bar{V}_{h+1}^\pi \left( s', b_{s'}^* \right) \\
&= \bar{V}_h^\pi(s, b) \\
&= \bar{V}_h^*(s, b).
\end{aligned}
$$

The first inequality used the fact that $(a^*, \hat{\mathbf{b}}^*) \in \hat{\mathcal{A}}_h(s, b)$. The second inequality follows from the induction hypothesis. The last two equalities follow from the standard policy evaluation equations and the definition of $\pi$, respectively. This completes the proof.

$\square$

### D.3. Proof of Lemma 5.3

*Proof.* We proceed by induction on $h$.

**Base Case.** For the base case, we consider $h = H + 1$. By definition and assumption, $\hat{V}_{H+1}^\pi(s, \hat{b}) = -\chi_{\{\hat{b} \geq 0\}} > -\infty$. Thus, it must be the case that $\hat{b} \geq 0$ and so by definition $\hat{C}_{H+1}^\pi(s, \hat{b}) = 0 \leq \hat{b}$.

**Inductive Step.** For the inductive step, we consider any $h \leq H$. As in the proof of Lemma 3.3, we know that $(a, \hat{\mathbf{b}}) = \pi_h(s, b) \in \hat{\mathcal{A}}_h(s, \hat{b})$ and for any $s' \in \mathcal{S}$ with $P_h(s' \mid s, a) > 0$ that $\hat{V}_{h+1}^\pi(s', b_{s'}) > -\infty$. Thus, the induction hypothesis implies that $\hat{C}_{h+1}^\pi(s', \hat{b}_{s'}) \leq \hat{b}_{s'} + \ell(S+1)(H-h)$ for any such $s'$. For any other $s'$, we have $g(P_h(s' \mid s, a)) = g(0) = 0$ by assumption.

Thus, the weighted combination of $\hat{C}_{h+1}^\pi(s', \hat{b}_{s'})$ is equal to the weighted combination of $\hat{\mathbf{b}}'$ where $\hat{b}_{s'}' \stackrel{\text{def}}{=} \hat{C}_{h+1}^\pi(s', \hat{b}_{s'})$ if

$P_h(s' \mid s, a) > 0$ and $\hat{b}'_{s'} \overset{\text{def}}{=} 0$ otherwise. Moreover, we have $\hat{b}' \leq \hat{b} + \ell(S+1)(H-h)$ since $\ell > 0$. Thus, by (SR),

$$
\begin{aligned}
\hat{C}_h^\pi(s, \hat{b}) &= c_h(s, a) + \underset{s'}{f}\, g(P_h(s' \mid s, a))\hat{C}_{h+1}^\pi(s', \hat{b}_{s'}) \\
&= c_h(s, a) + f_{h,\hat{\mathbf{b}}'}^{s,a}(1, 0) \\
&\leq c_h(s, a) + f_{h,\hat{\mathbf{b}}}^{s,a}(1, 0) + \ell(S+1)(H-h) \\
&\leq \kappa(\hat{b}) + \ell(S+1)(H-h) \\
&= \hat{b} + \ell(S+1)(H-h+1).
\end{aligned}
$$

The first inequality used Lemma D.1. The second inequality used the fact that $(a, \hat{\mathbf{b}}) \in \mathcal{A}_h(s, \hat{b})$. The last line used the definition of $\kappa$. This completes the proof. □

## D.4. Proof of Theorem 5.4

*Proof.* If (CON) is feasible, then inductively we see that $\hat{V}_1^*(s_0, \lceil B \rceil_\ell) > -\infty$. The contrapositive then implies if $\hat{V}_1^*(s_0, \lceil B \rceil_\ell) = -\infty$, then (CON) is infeasible. Thus, when Algorithm 4 outputs "Infeasible", it is correct.

On the other hand, suppose $\hat{V}_1^*(s_0, \lceil B \rceil_\ell) > -\infty$ and that $\pi$ is an optimal solution to $\hat{M}$. By Lemma 5.2 and Lemma 3.2, we know that $\hat{V}_1^\pi(s_0, \lceil B \rceil_\ell) \geq \bar{V}_1^\pi(s_0, B) \geq V^*$. Also, by Lemma 5.3, we know that $\hat{C}_1^\pi(s_0, \lceil B \rceil_\ell) \leq \lceil B \rceil_\ell + \ell(S+1)H \leq B + \ell(1 + (S+1)H)$. Our choice of $\ell = \frac{\epsilon}{1+(S+1)H}$ then implies that $\hat{C}^\pi = \hat{C}_1^\pi(s_0, \lceil B \rceil_\ell) \leq B + \epsilon$. Thus, $\pi$ is an $(0, \epsilon)$-additive bicriteria approximation for (CON).

Both cases together imply that Algorithm 4 is a valid $(0, \epsilon)$-bicriteria.

**Time Complexity.** We see immediately from Theorem 4.7 that the running time of Algorithm 4 is at most $O\left(H^{2m+1}S^{2m+2}A|\hat{\mathcal{B}}|^2 \|c_{max} - c_{min}\|_\infty^m / \epsilon^m\right)$. To complete the analysis, we need to bound $|\hat{\mathcal{B}}|$. First, we note $|\hat{\mathcal{B}}|$ is at most the number of integer multiples of $\ell$ in the range $[b_{min}, b_{max}] \subseteq [Hc_{min}, Hc_{max}]^m$. For any individual constraint, this number is at most $O(H(c_{max} - Hc_{min})/\ell) \leq O(H^2 S(c_{max} - c_{min})/\epsilon)$ using the definition of $\ell = \frac{\epsilon}{1+(S+1)H}$. Thus, the total number of rounded artificial budgets is at most $O((H^2 S \|c_{max} - c_{min}\| / \epsilon)^m)$. Squaring this quantity and plugging it back into our original formula yields: $O\left(H^{6m+1}S^{4m+2}A \|c_{max} - c_{min}\|_\infty^{3m} / \epsilon^{3m}\right)$.

□

## D.5. Proof of Proposition 5.8

*Proof.* We consider a reduction from the Hamiltonian Path problem. The transitions reflect the graph structure, and the actions determine the edge to follow next. To determine if a Hamiltonian path exists, we can simply make an indicator constraint for each node that signals that node has been reached. It is then clear that relaxing the budget constraint does not help since we can always shrink the budget for any given $\epsilon$-slackness. Thus, the claim holds. □

## D.6. Proof of Lemma 5.11

*Proof.* We proceed by induction on $h$.

**Base Case.** For the base case, we consider $h = H + 1$. By definition, we have $\tilde{V}_{H+1}^\pi(\tilde{\tau}_{H+1}) = 0 = V_{H+1}^\pi(\tau_{H+1})$ and $\tilde{C}_{H+1}^\pi(\tilde{\tau}_{H+1}) = 0 = C_{H+1}^\pi(\tau_{H+1})$.

**Inductive Step.** For the inductive step, we consider any $h \leq H$. For simplicity, let $x \stackrel{\text{def}}{=} \ell(\lambda_r + \lambda_p)Hr_{max}(s_{max} - s_{min})$. The standard policy evaluation equations imply that,

$$\tilde{V}_h^\pi(\tilde{\tau}_h) = r_h(\lceil s \rceil_\ell, a) + \sum_{\tilde{s}'} \tilde{P}_h(\tilde{s}' \mid \lceil s \rceil_\ell, a)\tilde{V}_{h+1}^\pi(\tilde{\tau}_{h+1})$$

$$= r_h(\lceil s \rceil_\ell, a) + \sum_{\tilde{s}'} \int_{s'=\tilde{s}'}^{\tilde{s}'+\ell} P_h(s' \mid \lceil s \rceil_\ell, a)ds'\tilde{V}_{h+1}^\pi(\tilde{\tau}_{h+1})$$

$$= r_h(\lceil s \rceil_\ell, a) + \int_{s'} P_h(s' \mid \lceil s \rceil_\ell, a)\tilde{V}_{h+1}^\pi(\tilde{\tau}_{h+1})ds'$$

$$\geq r_h(\lceil s \rceil_\ell, a) + \int_{s'} P_h(s' \mid \lceil s \rceil_\ell, a)(V_{h+1}^\pi(\tau_{h+1}) - x(H-h))ds'$$

$$= r_h(\lceil s \rceil_\ell, a) + \int_{s'} P_h(s' \mid \lceil s \rceil_\ell, a)V_{h+1}^\pi(\tau_{h+1})ds' - x(H-h)$$

$$\geq r_h(s, a) - \ell\lambda_r + \int_{s'} (P_h(s' \mid s, a) - \ell\lambda_p)V_{h+1}^\pi(\tau_{h+1})ds' - x(H-h)$$

$$= V_h^\pi(\tau_h) - \ell\lambda_r - \ell\lambda_p \int_{s'} V_{h+1}^\pi(\tau_{h+1})ds' - x(H-h)$$

$$\geq V_h^\pi(\tau_h) - \ell\lambda_r - \ell\lambda_p Hr_{max}(s_{max} - s_{min}) - x(H-h)$$

$$\geq V_h^\pi(\tau_h) - \ell(\lambda_r + \lambda_p)Hr_{max}(s_{max} - s_{min}) - x(H-h)$$

$$= V_h^\pi(\tau_h) - x(H - h + 1).$$

If we let $y \stackrel{\text{def}}{=} \ell(\lambda_c + \lambda_p)Hc_{max}(s_{max} - s_{min})$, we also see that,

$$\tilde{C}_h^\pi(\tilde{\tau}_h) = c_h(\lceil s \rceil_\ell, a) + \tilde{f}_{\tilde{s}'}\tilde{P}_h(\tilde{s}' \mid \lceil s \rceil_\ell, a)\tilde{C}_{h+1}^\pi(\tilde{\tau}_{h+1})$$

$$= c_h(\lceil s \rceil_\ell, a) + \tilde{f}_{\tilde{s}'} \int_{s'=\tilde{s}'}^{\tilde{s}'+\ell} P_h(s' \mid \lceil s \rceil_\ell, a)ds'\tilde{C}_{h+1}^\pi(\tilde{\tau}_{h+1})$$

$$= c_h(\lceil s \rceil_\ell, a) + \underset{s'}{f} P_h(s' \mid \lceil s \rceil_\ell, a)\tilde{C}_{h+1}^\pi(\tilde{\tau}_{h+1})$$

$$\leq c_h(\lceil s \rceil_\ell, a) + \underset{s'}{f} P_h(s' \mid \lceil s \rceil_\ell, a)(C_{h+1}^\pi(\tau_{h+1}) + y(H-h))$$

$$\leq c_h(\lceil s \rceil_\ell, a) + \underset{s'}{f} P_h(s' \mid \lceil s \rceil_\ell, a)C_{h+1}^\pi(\tau_{h+1}) + y(H-h)$$

$$\leq c_h(s, a) + \ell\lambda_c + \underset{s'}{f} (P_h(s' \mid s, a) + \ell\lambda_p)C_{h+1}^\pi(\tau_{h+1}) + y(H-h)$$

$$= c_h(s, a) + \underset{s'}{f} P_h(s' \mid s, a)C_{h+1}^\pi(\tau_{h+1}) + \ell\lambda_c + \ell\lambda_p \underset{s'}{f} C_{h+1}^\pi(\tau_{h+1}) + y(H-h)$$

$$\leq C_h^\pi(\tau_h) + \ell\lambda_c + \ell\lambda_p \underset{s'}{f} Hc_{max} + y(H-h)$$

$$\leq C_h^\pi(\tau_h) + \ell\lambda_c + \ell\lambda_p(s_{max} - s_{min})Hc_{max} + y(H-h)$$

$$\leq C_h^\pi(\tau_h) + \ell(\lambda_c + \lambda_p)(s_{max} - s_{min})Hc_{max} + y(H-h)$$

$$= C_h^\pi(\tau_h) + y(H - h + 1).$$

We note the above also holds if $P$ is replaced with a $g(P)$ for a sublinear short map $g$.

For almost-sure constraints, the proof is slightly different since we need to keep the inner integral by definition of the

worst-case cost for continuous state spaces. Letting $y \stackrel{\text{def}}{=} \ell(\lambda_c + \lambda_p)Hc_{max}(s_{max} - s_{min})/\tilde{p}_{min}$, the bound then becomes,

$$\tilde{C}_h^\pi(\tilde{\tau}_h) = c_h(\lceil s \rceil_\ell, a) + \max_{\tilde{s}'}[\tilde{P}_h(\tilde{s}' \mid \lceil s \rceil_\ell, a) > 0]\tilde{C}_{h+1}^\pi(\tilde{\tau}_{h+1})$$

$$= c_h(\lceil s \rceil_\ell, a) + \max_{\tilde{s}'}[\int_{s'=\tilde{s}'}^{\tilde{s}'+\ell} P_h(s' \mid \lceil s \rceil_\ell, a)ds' > 0]\tilde{C}_{h+1}^\pi(\tilde{\tau}_{h+1})$$

$$= c_h(\lceil s \rceil_\ell, a) + \max_{\tilde{s}'} \frac{\int_{s'=\tilde{s}'}^{\tilde{s}'+\ell} P_h(s' \mid \lceil s \rceil_\ell, a)ds'}{p_{\tilde{s}'}}\tilde{C}_{h+1}^\pi(\tilde{\tau}_{h+1})$$

$$= c_h(\lceil s \rceil_\ell, a) + \max_{\tilde{s}'} \int_{s'=\tilde{s}'}^{\tilde{s}'+\ell} P_h(s' \mid \lceil s \rceil_\ell, a)\tilde{C}_{h+1}^\pi(\tilde{\tau}_{h+1})ds'/p_{\tilde{s}'}$$

$$\leq c_h(\lceil s \rceil_\ell, a) + \max_{\tilde{s}'} \int_{s'=\tilde{s}'}^{\tilde{s}'+\ell} P_h(s' \mid \lceil s \rceil_\ell, a)(C_{h+1}^\pi(\tau_{h+1}) + y(H - h))ds'/p_{\tilde{s}'}$$

$$\leq c_h(\lceil s \rceil_\ell, a) + \max_{\tilde{s}'} \int_{s'=\tilde{s}'}^{\tilde{s}'+\ell} P_h(s' \mid \lceil s \rceil_\ell, a)C_{h+1}^\pi(\tau_{h+1})ds'/p_{\tilde{s}'} + y(H - h)$$

$$\leq c_h(s, a) + \ell\lambda_c + \max_{\tilde{s}'} \int_{s'=\tilde{s}'}^{\tilde{s}'+\ell} P_h(s' \mid s, a)C_{h+1}^\pi(\tau_{h+1})ds'/p_{\tilde{s}'}$$

$$+ \ell\lambda_p \max_{\tilde{s}'} \int_{s'=\tilde{s}'}^{\tilde{s}'+\ell} C_{h+1}^\pi(\tau_{h+1})ds'/p\tilde{s}' + y(H - h)$$

$$\leq c_h(s, a) + \max_{\mathcal{S}'\subseteq\mathcal{S}} \int_{\mathcal{S}'} \frac{P_h(s' \mid s, a)}{p_{\mathcal{S}'}}C_{h+1}^\pi(\tau_{h+1})ds' + \ell\lambda_c + \ell^2\lambda_p Hc_{max}/\tilde{p}_{min}$$

$$+ y(H - h)$$

$$= C_h^\pi(\tau_h) + \ell\lambda_c + \ell^2\lambda_p Hc_{max}/\tilde{p}_{min} + y(H - h)$$

$$\leq C_h^\pi(\tau_h) + \ell(\lambda_c + \lambda_p)(s_{max} - s_{min})Hc_{max}/\tilde{p}_{min} + y(H - h)$$

$$= C_h^\pi(\tau_h) + y(H - h + 1).$$

$\square$

## D.7. Proof of Theorem 5.12

*Proof.* The theorem follows immediately from Theorem 5.4 and Lemma 5.11. $\square$

# E. Extensions

**Markov Games.** It is easy to see that our augmented approach works to compute constrained equilibria. For efficient algorithms, using $-\infty$ to indicate infeasibility becomes problematic. However, we can still use per-stage LP solutions and add a constraint that the equilibrium value must be larger than some very small constant to rule out invalid $-\infty$ solutions. Alternatively, the AND/OR tree approach used in (McMahan and Zhu, 2024a) can be applied here to directly compute all the near-feasible states.

**Infinite Discounting.** Since we focus on approximation algorithms, the infinite discounted case can be immediately handled by using the idea of effective horizon. We can treat the problem as a finite horizon problem where the finite horizon $H$ is defined so that $\sum_{h=H}^\infty \gamma^{h-1}c_{max} \leq \epsilon'$. By choosing $\epsilon'$ and $\epsilon$ small enough, we can get equivalent feasibility approximations. The discounting also ensures the effective horizon $H$ is polynomially sized, implying efficient computation.

**Stochastic Policies.** For stochastic policies, our approximate results follow from simply replacing each $\max_a$ and $\max_{b_t}$ with a general linear program over a finite distribution, which can be solved in polynomial time.

**Stochastic Costs.** For finitely-supported cost distributions, all results remain the same except for almost-sure/anytime constraints, which now must be written in the form:

$$C_h^\pi(\tau_h) = \max_{c \in \text{Supp}(C_h(s,a))} c + \max_{s'}[P_h(s' \mid s, a) > 0] C_h^\pi(\tau_h, a, c, s'). \tag{20}$$

Also, note that histories must now be cost-dependent.

Now, we have that future budgets depend on both the next state and the realized cost, so our (ADP) must now be dependent on both states and immediate costs for subproblems. The construction is similar to the approach in (McMahan, 2024).

