# OpenReview forum: "Polynomial-Time Approximability of Constrained Reinforcement Learning"
_ICML.cc/2025/Conference — ICML 2025 poster_

### Official Review · Reviewer_TjVX · 2025-02-17

**Overall Recommendation:** 5

**Summary:**

The authors study the computational complexity of constrained MDPs.
They show some novel results regarding finding approximately optimal policies.

In particular, they studied the question of whether polynomial time approximation algorithms exist for many of the classic formulations studied in the CRL literature.
They found that for the vast majority of constraints, including all the standard constraints, polynomial-time approximability is possible.
Their work resolves the polynomial-time approximability of many settings, some of which have been lacking in any polynomial-time algorithm.

**Claims And Evidence:**

Yes.

**Essential References Not Discussed:**

No.

**Experimental Designs Or Analyses:**

Yes, at a high level.

**Methods And Evaluation Criteria:**

Yes.

**Other Comments Or Suggestions:**

Please add an Impact Statement!

**Other Strengths And Weaknesses:**

This work is to the best of my knowledge, original.

The authors are the first to prove polynomial-time approximability for the following settings: policies under chance constraints, deterministic policies under multiple expectation constraints, policies under non-homogeneous constraints (i.e., constraints of different types), and policies under constraints for continuous-state processes.
This fills an important gap in the literature, and for that matter I am vouching for this paper to be accepted.

**Questions For Authors:**

Page 1:
Please frame your main question in a more general way.

Page 2:
Could you please add some discussion on CAMDPs?

Page 3:
I do not understand Definition 2.4.

Page 3:
Can you please add some comments regarding Proposition 2.3 (proof sketch)?
I understand that the proof is in the appendix, but some justification must be added in the main body as well.

Page 4:
In Equation (2), did you perhaps mean to sum over $a$?

Please further explain Equation (4).

Page 5:
Please elaborate on the functionalities of Algorithm 1 and Algorithm 2, in their respective captions.

Page 6:
I do not understand Lines 302 to 309 (second column).

Page 7:
Theorem 5.4:
Why is this running time polynomial?

Page 8:
Please further elaborate on future work.

**Relation To Broader Scientific Literature:**

Contributions are relevant and important for the broader RL community, as they prove long-sought guarantees for various types of RL tasks, such as polynomial-time approximability for policies under chance constraints, deterministic policies under multiple expectation constraints, policies under non-homogeneous constraints (i.e., constraints of different types), and so on.

**Theoretical Claims:**

I only checked the main body :)

---

> ### Author Rebuttal · Authors · 2025-03-31
>
> Thank you for your constructive feedback! We will update the paper to clarify the points the reviewer made. To address your questions,
>
> **[Page 4]** This is meant to be s' as is. Note, this equation corresponds to the usual policy evaluation equations for a deterministic policy. Here, we are assuming a is the specific action chosen by the policy.
>
> **[Page 7]** Note, each quantity involved is part of the MDP, so smaller than its description size. Recall that m, the number of constraints, must be constant or else the problem cannot be solved efficiently. Given m is constant, quantities such as S^m are then polynomial-sized. For example, given only one constraint, the bound is $O(H^7 S^6 A poly(|M|)/\epsilon^3)$, which can be seen to be polynomial in $|M|$ and $1/\epsilon$.
>
> Thanks again and let us know if there is anything else that needs clarifying!

---

### Official Review · Reviewer_c1yJ · 2025-03-10

**Overall Recommendation:** 3

**Summary:**

The paper develops approximation algorithms for constrained MDP problems. The problem formulation in the paper can incorporate a broad class of recursively computable constraints, including almost-sure, chance, expectation, and their anytime variants. For this general problem formulation, the paper develops a framework to compute a near-optimal policy that achieves the same objective value as the optimal policy, while ensuring that the constraints are satisfied with an additive error $\epsilon$.

The general approach developed in the paper is based on an augmentation scheme. To be specific, the paper chooses a future artificial budget to replace the original constraints. The artificial budget imposes some restrictions on the actions that can be taken at each round, and then enables us to reduce the original constrained MDP problems into MDP problems. Then, with this reduced MDP problem, the paper can develop the associated Bellman updates and employs a dynamic programming to solve the problem. However, the dynamic programming can be hard to solve, the paper then develops a dynamic rounding approach to ease the computation. Finally, with an additional approximation method that approximates the possible set of artificial budget with a finite set, the paper develops an $\epsilon$-approximate policy with a computation time polynomial in $1/\epsilon$. The paper further extends their approaches to continuous settings, as well as function approximation settings.

**Claims And Evidence:**

The claim is clear and looks convincing.

**Essential References Not Discussed:**

Most papers are discussed.

**Experimental Designs Or Analyses:**

There is no numerical experiment in the paper.

**Methods And Evaluation Criteria:**

The method makes sense, however, there is no numerical validation of their algorithms, as I further explain in the strength and weakness part.

**Other Comments Or Suggestions:**

1. In Section 6, it is said that an OpenAI gym environment is developed to model the approach with function approximation on this CMDP. However, I cannot find further details in the paper.

2. There is no impact statement in the paper.

**Other Strengths And Weaknesses:**

Strength:

The paper develops a general approach to handle the CMDP problems, where the constraints can incorporate many formulation, for example almost surely, chance, expectation, and the anytime variants.

Weakness:

1. The comparison with the previous literature can be made better. The CMDP problems are classical problems, and many algorithms have been developed to solve the CMDP problems. However, it is not clear from the paper whether the computation time bound improves over the previous results.

2. The writing of the paper could be further improved, especially the descriptions of the methods in Section 6.

3. There is no numerical experiment over the algorithms developed in the paper.

4. The constrained MDP problems considered in the paper actually assume that all the problem parameters are given and the algorithms in the paper rely heavily on the prior knowledge of the cost, as well as transition probabilities. Therefore, I do not see any learning element in the paper, though the title of the paper is about reinforcement learning.

**Questions For Authors:**

1. It seems to me that your algorithm will always produce a deterministic policy. However, it is well known that the optimal policy for a CMDP problem can be a randomized policy, especially when the constraint takes an expectation formulation. Is it possible that your algorithm will output a randomized policy?

2. Since the paper assumes that all the problem parameters are given, could we just formulate the CMDP problem as an optimization problem, and then use the existing optimization algorithm to solve it? For example, Altman (1999) just formulates the CMDP problems as linear programming to solve it.

3. How does the computation time bound developed in the paper compare to previous bounds established in the literature?

4. Since the title of the paper is regarding reinforcement learning, how to implement the algorithm when the problem parameters are not given, for example, the cost and the transition probabilities are unknown.

5. Could you comment on the real performance of your algorithm?

**Relation To Broader Scientific Literature:**

The main contribution of the paper is to develop a general method to approximate the CMDP problems, where the constraints can take many formulations, including almost-sure, chance, expectation, and anytime variants. The paper computes a policy that enjoys the same objective value as the optimal policy, and violates the constraint up to an additive $\epsilon$ bound, with a computation time polynomial in $1/\epsilon$.

**Theoretical Claims:**

I have checked the proofs and the theoretical claims. They look correct to me.

---

> ### Author Rebuttal · Authors · 2025-03-31
>
> Thank you for your constructive feedback! Please see our general rebuttal above in Reviewer htBq's rebuttal section, which addresses Weaknesses 2-4 and other points the reviewer brought up. In addition to those points, we address several points below.
>
> **[Weakness 1]** In our paragraph "Past Work" in the introduction, we explicitly mentioned all known works that achieve polynomial-time approximation algorithms for special cases of our problem. Are there any specific works the reviewer is thinking of that we have missed here? All other works we are aware of are exponential time or apply to completely different settings, and thus are immediately inferior or incomparable to our method within the context of establishing polynomial-time approximability of our problem. However, we are happy to include references to exponential time algorithms if the reviewer feels this will make connections to the literature clearer.
>
> **[Q1]** Note, in the appendix, we show how to modify our method to achieve stochastic policies. However, we emphasize that the computation of stochastic, expectation-constrained policies is already known and so not a goal of our paper. Our goal is to compute deterministic, expectation-constrained policies and policies under other constraints, which are problems that have yet to be solved in the literature.
>
> **[Q2]** Note, this approach only works for *stochastic, expectation-constrained policies*. The LP formulation no longer holds when policies are additionally constrained to be deterministic or are forced to obey more complex anytime or chance constraints. This is immediate since our problem is NP-hard as mentioned in the "Hardness" paragraph, so any such polynomial-sized LP formulation would immediately imply that P = NP.
>
> **[Q3]** As in our Weakness 1 discussion, we again emphasize that the only goal of our paper is to devise *polynomial-time* algorithms for problems that currently have none. As such, the only distinction we focus on is polynomial runtimes vs exponential runtimes. Since all the open problems we discuss either have only exponential time algorithms or have no algorithm whatsoever, our algorithms beat all such competitors by definition in the context of computational complexity. Again, we are happy to include references to exponential time algorithms if the reviewer feels this will make connections to the literature clearer.
>
>
> **[Q4 + Q5]** Please see our general rebuttal in addition to the discussion above for Q3.
>
> Thank you again for your feedback, and let us know if there is anything else we can clarify!

---

### Official Review · Reviewer_p12Q · 2025-03-12

**Overall Recommendation:** 3

**Summary:**

The submission provides a novel generic bicriteria approximation algorithm that covers a broad range of Constrained Markov Decision Processes (CMDPs).

**Claims And Evidence:**

Yes, with the exception of the malformed definitions and anecdotal experimental setup (see "Other Strengths and Weaknesses").

**Essential References Not Discussed:**

N/A.

**Experimental Designs Or Analyses:**

N/A.

**Methods And Evaluation Criteria:**

The submission is of foundational nature and the methods (proofs) are appropriate.

**Other Comments Or Suggestions:**

-Typo on page 2: "bicritiera" should be "bicriteria", and there should not be an "a" on row "64".
-The running title still has the placeholder "Submission and Formatting Instructions for ICML 2025".
-Row 196: "Our revolves around" - I suggest rephrasing, as it is not clear what "Our" refers to.
-Row 400: "The user also have" -> "... has"

**Other Strengths And Weaknesses:**

On the positive side, the overall contribution of the submission is solid: while it lacks experimental evaluations (contrasting many of the preceding papers), it does provide a non-trivial theoretical contribution which pushes the boundaries of approximability for a range of CMDPs. I am not an expert on CMDPs, but the description of the related work to me convincingly justified the relevance of these problems.

The inclusion of the to-the-point "Our Techniques" paragraph is a nice touch that is often missing from theoretical ICML submissions. Nevertheless, given how close the result is to the recently developed polynomial-time (0, ϵ)-additive bicriteria approximation algorithms of McMahan and Zhu (AISTATS 2024), I would have expected a closer treatment of how the techniques differ from those used previously. There is a brief note on this at the start of Section 3, but does that paragraph imply that there is no intersection between the techniques used here and those employed in the preceding AISTATS 2024 paper?

Regarding experimental evaluations, the second-to-last paragraph of the submission mentions a seemingly anecdotal experimental setup without providing any proper details. I believe that in the current form, these claims should not appear in the main body of the submission as they are entirely non-reproducible; they can either be expanded to a proper experimental setup (with details), or moved entirely to the appendix. As the submission's contribution is foundational, I do not consider the lack of experiments to be a significant issue.

-On the negative side, the problem is not properly formalized, or more precisely, the provided definition in Section 2 is malformed. It defines a caMDP as a tuple of the form (H,S,...), which is fine - however, when describing the elements of the tuple, it never speaks about S but rather about S_h where h is never defined or explained (and similarly for A_h etc.). Moreover, is the "time horizon" H an integer? What is the triangle(S) and triangle(R) notation? And are s,a elements of S and A, respectively? These things have to be made crystal clear, as without the formal definitions the vast majority of the ICML audience would not be able to understand the specific formal model used (and hence the contributions become meaningless). Due to these issues, I could not verify the formal correctness of the proofs.

Formal issues aside, given the complexity of the problem formulation it would be natural to include a simple example of a concrete problem instance (possibly highlighting various types of constraints). This could be, e.g., a running example that guides readers through Section 2, or just some summarizing examples at the end of that section.

## Update after rebuttal
Thank you for the responses; I maintain my assessment and score.

**Questions For Authors:**

The authors are welcome to respond to any of the points raised in the "Other Strengths And Weaknesses" section, and in particular clarify the problem definition while providing some concrete example of a problem instance. I am open to adjusting my score, but cannot currently recommend acceptance given the current state of the presentation.

On a related note, is there a reason for using CMDP in some places, cMDP in others and caMDP elsewhere?  Are these the same things merely using inconsistent abbreviations? As far as I could see, only caMDP is formally defined in the paper.

**Relation To Broader Scientific Literature:**

The results build on and extend previous works on solving CMDPs; the most closely related paper is the AISTATS 2024 paper which provides a bicriteria approximation algorithm for a subset of the instances considered here.

**Theoretical Claims:**

Only some, due to issues with malformed definitions.

---

> ### Author Rebuttal · Authors · 2025-03-31
>
> Thank you for your constructive feedback! Please see our general rebuttal above in Reviewer htBq's rebuttal section.
>
> **[Formalization]** In addition, we address the claim that our problem is not properly formalized. As our work is within the domain of CRL, we assume some basic familiarity with CRL notations and conventions. Specifically, our definition of a caMDP is almost identical to the usual definition of a "constrained MDP" used in other CRL works, including the AISTATS paper [1] that the reviewer referenced. That being said, we are happy to break the norms of the literature to enhance the clarity of the paper. To this end, we will add the following clarifications:
> 1) H is a positive integer,
> 2) s represents a state,
> 3) a represents an action,
> 4)  $\Delta(X)$ represents the probability simplex over elements of a set X (so $\Delta(S)$ is the set of all possible distributions over states), and
> 5) Lowercase c denotes the cost function when costs are deterministic, similar to lowercase r.
>
> **[Non-Stationarity]** To clarify to the reviewer, we used $S_h$ and $A_h$ in the definition of caMDPs because Definition 3.1 requires a non-stationary state and action space that may change over time, similar to Definition 2 in [1]. To clarify this, we will revert to a classic stationary state space in the caMDP definition, and instead define the non-stationary state and action space concept more explicitly in Definition 3.1, where that concept is first needed. Consequently, the caMDP definition would then take the more standard form $(S, A, H, P, R, C, s_0)$.
>
> **[CMDP Definition]** We also would like to point out that we have formally defined a CMDP in the paragraph "Constrained Processes" to be the tuple $(M, C, B)$ consisting of a caMDP along with a cost criterion $\mathcal{C}$ and a budget B. We introduce this definition because known works on constrained MDPs only consider a single constraint type, so the cost criterion is usually assumed from context. However, since we consider many different types of constraints here, we felt it would be useful to explicitly specify the cost criterion in the environment's definition. If it would be clearer, we are also happy to skip the definition of a caMDP and define a CMPD as the 9-tuple: $(S, A, H, P, R, C, s_0, \mathcal{C}, B)$. Note, the lowercase cMDP is just a typo that should be CMDP. As mentioned above, CMDP and caMDP are different.
>
> **[Comparison to [1]]** Concerning a comparison to [1], we bring the reviewer's attention to the section 3 statement: "When moving to other constraints, the cumulative cost may no longer suffice to determine constraint satisfaction. For example, the expected cost depends on the cumulative cost
> of all realizable branches, not just the current branch." Thus, unlike the simple augmentation of cumulative cost as done in [1], here we must use a more abstract notion of artificial future budgets with our state augmentation. This results in an exponentially sized augmented action space that cannot be solved efficiently using standard algorithms, unlike the simpler MDP constructed in [1] without action augmentation. This forces us to design an entirely new approximate dynamic programming (DP) approach, which makes up the entirety of section 4. Lastly, in the paragraph  "time-space errors", we mentioned that we use a customized rounding scheme due to errors accumulating over space and time in our approximate DP approach, unlike that of [1]'s projection scheme. These are all key fundamental differences needed to tackle combinations of constraints beyond anytime constraints.
>
> **[Running Example]** We will also add a self-driving car example commonly used in CRL. Specifically, we wish to construct a route that minimizes time for the car to reach its destination while obeying 1) an anytime constraint on its gas consumption so that it does not run out of fuel, 2) an expectation constraint on its CO2 consumption to minimize damage to the Earth, and 3) a chance constraint ensuring the car has a low probability of sustaining damage on the route.
>
> Please let us know if there are any other items we can clarify, and thank you again for your feedback!

---

> > ### Comment · Reviewer_p12Q · 2025-04-02
> >
> > Regarding the sentence "As our work is within the domain of CRL, we assume some basic familiarity with CRL notations and conventions.",
> > I think this is a bit of an unfortunate approach that, if adopted more widely, would completely defeat the purpose of ICML as a broad conference connecting distinct communities working on ML. For instance, when writing ICML papers, researchers working more on the complexity-theoretic side of things do introduce basic tools (such as the exponential time hypothesis or treewidth) even if these are well-known by everyone in their research community. The same holds for submissions that provide new contributions to Bayesian networks, constraint satisfaction, teaching dimension and so forth; a submission to ICML would generally focus much more on approachability than a submission to a specialized venue.
> >
> > Nevertheless, if the authors are willing to expand on the definitions to avoid assuming "familiarity with CRL notations and conventions", add an example and remove the informal references to experiments (as per their response to Reviewer htBq), I will not push for rejection and have updated my score accordingly.

---

### Official Review · Reviewer_htBq · 2025-03-14

**Overall Recommendation:** 3

**Summary:**

In this paper, they study polynomial algorithm for a generalized constrained MDP (CMDP).
In particular, they formulate a class of CMDPs whose constraints have generalized Bellman-recursive structures, and
construct polynomial algorithms for it via state/action augmentation and budget/state discretization.
Moreover, they extend their method by replacing the augmented component with function approximation.
Finally, they show an application of the proposed method as a fast, approximate solver for knapsack problem.

**Claims And Evidence:**

Reinforcement learning is a subfield of machine learning, which I believe involves some form of inference from finite sample. However, the presented results assume the complete knowledge of CMDP. Therefore I think the title is more accurate with something like "Polynomial-Time Solvability of Constrained Markov Decision Process".

**Essential References Not Discussed:**

None that I am aware of.

**Experimental Designs Or Analyses:**

The presentation of Knapsack Example could be more detailed in a quantitative aspect.

**Methods And Evaluation Criteria:**

The presentation of Knapsack Example could be more detailed in a quantitative aspect.

**Other Comments Or Suggestions:**

* L074 left: "SR" appears here for the first time without explanation.
* L097 left (and more): decide which one to use: cMDP or CMDP?
* L099 right: Give the total description size of caMDP explicitly.
* L133 left: $t$ starting from 1, but the initial time step is 0.
* L155 left: "finite extension of $f$" is ambiguous. I personally prefer explanations with mathematical expression, like $f_{t=1}^T x_t:=f(f(\cdots f(f(x_1, x_2), x_3), \cdots, x_{T-1}), x_T)$.
* L145 right: what is the truncated-horizon process?
* L196 left: Our *method(?)* revolves ...
* L262 left: Define $\bar{C}_h^\pi(s,b)$
* L316 left: $\kappa_\ell$
* L321 left: $g(P_h(s'|s,a),b_{s'}))$

**Other Strengths And Weaknesses:**

None.

**Questions For Authors:**

None.

**Relation To Broader Scientific Literature:**

I am not an expert in constrained MDP literature, so not sure how significant its contributions are.
On one side, the proposed framework is of great generality, which is nice, yet on the other side, the theoretical analysis seems rather straightforward and the authors indicate no particular use of the widened applicability beyond the good old knapsack problem.

**Theoretical Claims:**

The idea behind the proof is easy to follow.

---

> ### Author Rebuttal · Authors · 2025-03-31
>
> Thank you for your constructive feedback! We intend to revise the paper with your notational suggestions in mind. Please see our general rebuttal below, which addresses your concerns about our title, the significance of our results, and the role of our knapsack example:
>
> --- *General Rebuttal* ---
>
> "We address the common concerns of the reviewers, particularly those who may not be experts in the Constrained Reinforcement Learning (CRL) domain. These concerns include using the term "Reinforcement Learning" (RL) in our title and the significance of our contributions to the CRL literature. We also clarify the intention behind the final section's knapsack example, given the theoretical nature of our work.
>
> 1. **[Title]** First, it is a common convention in the RL literature to use "RL" to reference both planning (full MDP knowledge) and learning settings, as seen in many of the past CRL works cited in our paper, such as [1,2,3]. Moreover, planning algorithms can directly translate into learning algorithms through model-based learning approaches. However, we understand this might be misleading to some reviewers, given our paper's focus on computational complexity. We are open to changing the title if the reviewers believe this would be beneficial.
>
> 2. **[Contributions]** Second, we would like to try to convey the significance of our contributions even to non-experts in CRL through their foundational nature. Specifically, we note that the entire objective of CRL is to compute a value-maximizing policy that satisfies constraints. From a computational perspective, the fundamental question is whether such an (approximately) optimal policy is computable in polynomial time. We not only make progress on this question for nearly all constraints studied by the CRL literature, but also derive algorithms that achieve the provably best approximation guarantees possible under $P \neq NP$. Thus, our work is the first to establish polynomial-time approximability for each setting mentioned in the abstract and introduction, including the natural heterogeneous and chance constraint [3] settings. Before our work, all of these settings lacked sub-exponential time algorithms, or any algorithm, since their first introductions over 25 years ago. In fact, a significant paper appearing in Neurips 2024 [2] achieved similar results to ours, but only for the special case of problems with exactly one constraint and excluding chance constraints.
>
> 3. **[Knapsack Example]** Lastly, we emphasize that our paper is theoretical work with the singular focus of addressing: "Can general CMDPs be approximated in polynomial time?" Since this is a question about the worst-case computational complexity of an optimization problem, this question can only be answered using a mathematical proof. In particular, numerical experiments cannot establish computational complexity bounds and thus do not serve our primary objective. Consequently, our final section on function approximation was only intended to highlight the flexibility of our approach rather than stand as an experimental section. We see now this may have been confusing, and are happy to remove that section if it distracts from the paper's main message on the computational complexity of CRL.
>
> **References**:
>
> [1] "Anytime-Constrained Reinforcement Learning", Jeremy McMahan and Xiaojin Zhou. AISTATS 2024.
>
> [2] "Deterministic Policies for Constrained Reinforcement Learning in Polynomial Time", Jeremy McMahan. NeurIPS 2024.
>
> [3] "Risk-Constrained Reinforcement Learning with Percentile Risk Criteria", Yinlam Chow, Mohammad Ghavamzadeh, Lucas Janson, and Marco Pavone. JMLR 2017
>
> --- *End of general rebuttal* ---
>
> In addition to those points, we would like to address the reviewer's statement about the straightforwardness of the results. We note that many works have attempted to develop polynomial-time algorithms for our settings since they were first studied over 25 years ago, yet we are the first to achieve this feat. We believe this fact alone implies our approach is not insignificant. Moreover, if our approach were straightforward, we would argue this is an additional benefit. Specifically, this would mean that our method is not only provably exponentially faster than past approaches but also achieves optimal approximation guarantees without relying on complicated machinery such as large MILPs and exhaustive searches.
>
> Thank you once again for your valuable feedback! We hope our responses address your concerns and clarify the contributions of our work.

---

### Decision · Program_Chairs · 2025-05-01

**Decision:**

Accept (poster)

**Comment:**

All the Reviewers agreed that the paper meets the bar for acceptance at ICML. The initial concerns of the Reviewers were solved by the Authors during the rebuttals. Thus, I will propose acceptance.